# NMDA spikes mediate amplification of inputs in the rat piriform cortex

**Amit Kumar[1], Oded Schiff[1], Edi Barkai[2], Bartlett W Mel[3]\*, Alon Poleg-Polsky[4]\*, Jackie Schiller[1]\***

[1]Department of Physiology, The Rappaport Faculty of Medicine and Research Institute, Technion-Israel Institute of Technology, Haifa, Israel; [2]Department of Neurobiology, University of Haifa, Haifa, Israel; [3]Biomedical Engineering Department, University of Southern California, Los Angeles, United States; [4]Department of Physiology and Biophysics, University of Colorado School of Medicine, Aurora, United States

**Abstract** The piriform cortex (PCx) receives direct input from the olfactory bulb (OB) and is the brain's main station for odor recognition and memory. The transformation of the odor code from OB to PCx is profound: mitral and tufted cells in olfactory glomeruli respond to individual odorant molecules, whereas pyramidal neurons (PNs) in the PCx responds to multiple, apparently random combinations of activated glomeruli. How these 'discontinuous' receptive fields are formed from OB inputs remains unknown. Counter to the prevailing view that olfactory PNs sum their inputs passively, we show for the first time that NMDA spikes within individual dendrites can both amplify OB inputs and impose combination selectivity upon them, while their ability to compartmentalize voltage signals allows different dendrites to represent different odorant combinations. Thus, the 2-layer integrative behavior of olfactory PN dendrites provides a parsimonious account for the nonlinear remapping of the odor code from bulb to cortex.
DOI: https://doi.org/10.7554/eLife.38446.001

**\*For correspondence:**
mel@usc.edu (BWM);
alon.poleg-polsky@ucdenver.edu
(AP-P);
jackie@technion.ac.il (JS)

**Competing interests:** The authors declare that no competing interests exist.

## Introduction

The piriform cortex (PCx) is the main cortical station in olfactory processing. It receives direct odor information from the olfactory bulb, as well as contextual information from higher brain regions, and is thought to be the brain's primary site for odor discrimination and recognition (*Gottfried, 2010*).

Olfaction starts at the nasal epithelium where a single odor activates multiple odorant receptors (ORs). At the olfactory bulb, information from like ORs converge to ~1000 mirror symmetric pairs of glomeruli (*Mombaerts et al., 1996*; *Isaacson, 2010*). Thus, on either side of the brain an odor is represented as a distributed pattern of activation over ~1000 glomeruli, which together form a molecular map of the odor. Mitral and tufted (M/T) cells, the glomerular outputs, carry the olfactory signal next to the PCx via the lateral olfactory tract (LOT). LOT axons traverse layer 1 of the PCx and form synaptic contacts with the thin distal dendrites of pyramidal neurons and layer 1 interneurons (*Gottfried, 2010*; *Bekkers and Suzuki, 2013*).

Unlike other sensory cortices which are topographically organized, the connectivity scheme between the OB and PCx lacks any apparent spatial structure: individual LOT axons from the bulb terminate in broad overlapping swaths of the PCx (*Buonviso et al., 1991a*; *Buonviso et al., 1991b*; *Ojima et al., 1984*; *Miyamichi et al., 2011*) so that each glomerulus excites a widely dispersed and apparently randomly distributed population of PCx neurons. In turn, each pyramidal neuron in PCx receives synaptic input from roughly ~10% of the 1000 glomeruli, also apparently randomly sampled (*Miyamichi et al., 2011*; *Davison and Ehlers, 2011*; *Nagayama, 2010*; *Sosulski et al., 2011*; *Ghosh et al., 2011*; *Stettler and Axel, 2009*; *Soucy et al., 2009*). In keeping with the notion of

random convergence and divergence of OB axons in PCx, electrophysiological studies show that each odorant activates an apparently random population of from 3% to 15% of the neurons in layer 2 of the PCx at low concentration (*Stettler and Axel, 2009*).

Pyramidal neurons in the piriform cortex are the main integration units within which the discrete molecular information channels of the OB are combined to form 'odor objects', but the biophysical and circuit-level mechanisms that remap LOT inputs into the olfactory code in PCx remain poorly understood. Three features of the mapping of LOT excitation into pyramidal neuron activity in PCx are noteworthy, and in the context of the existing literature, lead to a conundrum:

1. Pyramidal neurons are driven by LOT inputs (*Bekkers and Suzuki, 2013*), even though LOT synaptic contacts onto PNs are formed on distal tuft dendrites, are few in number (~200 total contacts, *Miyamichi et al., 2011*), and are sparsely activated (just 1% of glomeruli activated in the OB reliably drives many PCx pyramidal neurons (*Davison and Ehlers, 2011*). This suggests pyramidal neurons in PCx have some means of amplifying weak distal inputs.
2. Pyramidal neurons in PCx are combination selective, that is, they respond supralinearly to specific combinations of glomerular inputs but not others (*Davison and Ehlers, 2011*). These combinations are of relatively high order, so that a pyramidal neuron that responds strongly to an odorant may respond weakly to a chemically similar odorant that activates a heavily overlapping pattern of glomeruli .
3. Pyramidal neurons in PCx have 'discontinuous' receptive fields, that is, they respond to multiple chemically diverse odorants (*Stettler and Axel, 2009*; *Rennaker et al., 2007*; *Kadohisa and Wilson, 2006*; *Poo and Isaacson, 2009*), while failing to respond to re-combinations of those odorants' component parts (*Davison and Ehlers, 2011*; *Apicella et al., 2010*). For example, a pyramidal neuron that is unresponsive to individual odor components A, B, or C may respond strongly to combinations AB but fail to respond to combination AC or BC.

What biophysical mechanism(s) could account for a pyramidal neuron's ability to (1) amplify distal LOT inputs; (2) enforce combination selectivity on those inputs; and (3) maintain multiple 'discontinuous' recognition subunits? A possible mechanism could be compartmentalized NMDA spikes in pyramidal neuron dendrites, which could provide both the thresholding nonlinearity that enforces combination selectivity, and the amplification that allows distal inputs to drive somatic action potentials (*Poirazi et al., 2003*; *Polsky et al., 2004*; *Larkum et al., 2009*; *Sheffield and Dombeck, 2015*; *Losonczy and Magee, 2006*; *Nevian et al., 2007*; *Milojkovic et al., 2004*).

Weighing against this hypothesis, however, the one study that has analyzed dendritic responses of olfactory PNs using current injections and focal electrical stimulation reported that pyramidal neurons in the PCx lack sufficient NMDA (or other) regenerative currents that could provide either the combination selectivity or amplification of LOT inputs (*Bathellier et al., 2009*). Rather, Bathellier et al. reported that pyramidal neurons in PCx, unlike their counterparts in other cortical areas, act as intrinsically linear summing units. This leads to a conundrum: the only alternative source of nonlinearity that would seem capable of producing combination selectivity – recurrent network effects mediated by intracortical inputs (IC) to pyramidal neurons – has also apparently been ruled out: *Apicella et al., 2010* tested whether a pyramidal neuron's ability to respond selectively to LOT input combinations depends on IC inputs, which outnumber a pyramidal neuron's LOT inputs 10 to 1, but they found no reduction in a pyramidal neuron's combination selectively when its IC inputs were blocked with baclofen.

Given the importance of understanding the cellular mechanisms underlying odor representation in PCx, we revisited the question as to whether pyramidal neuron dendrites in PCx can generate local spikes (*Larkum et al., 2009*; *Nevian et al., 2007*; *Lavzin et al., 2012*; *Major et al., 2008*; *Antic et al., 2010*; *Larkum and Nevian, 2008*; *Stuart and Spruston, 2015*). We found that robust NMDA spikes can indeed be generated in dendrites of PCx pyramidal neurons, both in layer 1a and layer two which receives direct LOT input, and using a model we show that these local spikes can effectively amplify clustered versus distributed LOT inputs forming the basis for a discontinuous receptive field. We also show that supralinear summation of LOT inputs is largely confined to a single dendrite, whereas nonlinear interactions of LOT inputs between dendrites are weak. These findings support the idea that a pyramidal neuron in PCx can represent multiple distinct glomerular combinations within its apical dendritic arbor, which fulfills the basic requirements for a discontinuous receptive field (*Stettler and Axel, 2009*). Finally, we show that interactions between LOT and IC

inputs are also nonlinear, a fact that will likely be important for understanding the recurrent pattern completion functions of the PCx.

## Results

### Glutamate uncaging evoked NMDA spikes in apical dendrites of PCx pyramidal neurons

To directly address the fundamental question of whether dendrites of pyramidal neurons in PCx can generate dendritic spikes we used focal glutamate uncaging (MNI-glutamate) to activate specific dendritic locations while recording the somatic voltage. Neurons were loaded with the calcium sensitive dye OBG-6F (200 μM) and CF633 (200 μM) to visualize the dendritic tree and perform calcium

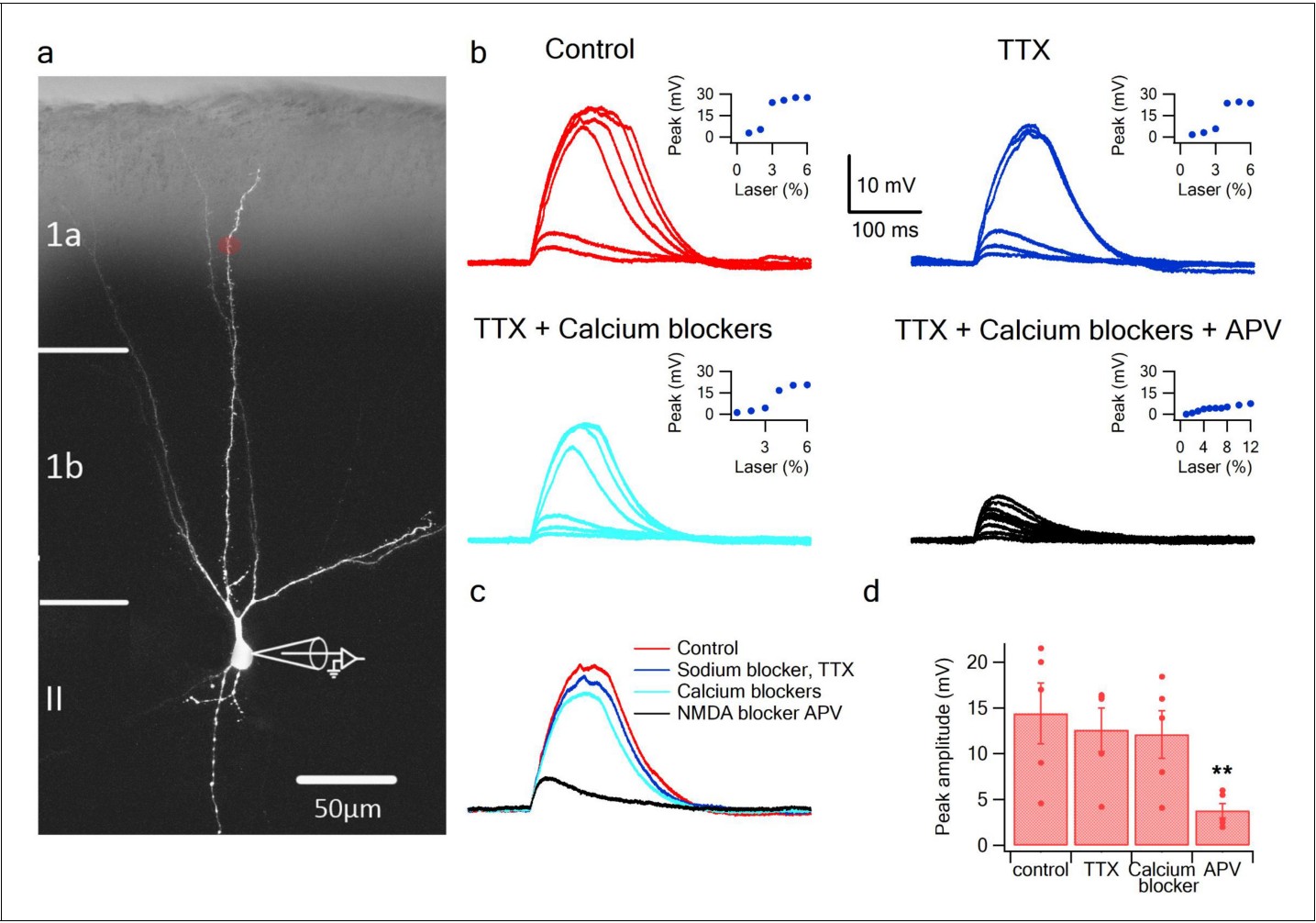

**Figure 1.** Glutamate uncaging evoked NMDA spikes in dendrites of PCx pyramidal neurons. (a) Fluorescence image reconstruction of a pyramidal neuron filled with CF633 (200 μM) via the patch recording electrode. Uncaging location is indicated by the red dot. (b) Voltage responses and dendritic spikes were evoked by uncaging of MNI-glutamate at increasing laser intensities in the Control condition (red); in the presence of the voltage gated sodium channel blocker TTX (1 μm; blue); with an additional cocktail of voltage-gated calcium channel blockers (w-agatoxin 0.5 μM, conotoxin-GVIA 5 μM; SNX 482 200 nM, nifedipine 10 μM; cyan), and finally adding APV (50 μm; black). Insets, peak voltage response at increasing laser intensity for the different conditions. (c) Overlay of the spike in the Control condition (red), in TTX (blue), in TTX + calcium channel blockers (cyan) and with APV (black). All traces were collected using the same laser stimulation intensity. The spike was completely blocked with APV, and could not be reinitiated at higher laser intensities. (d) Summary plot of the peak response amplitude in control, TTX, $Ca^{2+}$ blockers, and APV (n = 5). ** ANOVA test for comparison of all four groups (control, TTX, Ca blockers and APV) showed statistical significance (p < 0.05). Post-hoc analysis (Dunnett test) comparing each of the three groups (TTX, Ca blockers and APV) to the control yielded a significant difference only for comparison of APV to the control (p < 0.05).

DOI: https://doi.org/10.7554/eLife.38446.002

imaging (*Figure 1a*). The majority of our recorded neurons were pyramidal neurons from layer 2, as determined from the Dodt contrast image and somatic firing pattern (*Suzuki and Bekkers, 2006*). Gradually increasing the laser intensity to uncage glutamate evoked EPSP-like potentials, which increased progressively up to a threshold laser activation, beyond which a local spike was initiated (*Figure 1a–b*; n = 10 cells). We then perfused the slices with specific blockers for voltage-gated sodium, calcium and NMDAR channels (*Major et al., 2008*; *Schiller et al., 2000*). Addition of the voltage-gated sodium channel blocker TTX (1 μm) did not significantly change the peak of the slow spikes (90.8 ± 6.0% of control; p = 0.16; n = 5). In some cases, we recorded a fast spikelet that preceded the prolonged voltage plateau, reminiscent of the sodium spikelets observed in CA1 pyramidal dendrites and tuft dendrites of neocortical layer 5 pyramidal neurons (*Larkum et al., 2009*; *Ariav et al., 2003*); we verified that the fast spikelet was blocked by TTX (an example of a fast spikelet is shown in *Figure 2f* and Figure 4b). An added cocktail of voltage-gated calcium channel blockers (w-agatoxin 0.5 μM, conotoxin-GVIA 5 μM; SNX 482 200 nM, nifedipine10 μM) had a small, nonsignificant effect on the peak of the dendritic spike (85.4 ± 4.2% of control; p = 0.08; n = 5). Finally, APV (2-amino 5-phosphonovalerate; 50 μM) a specific NMDAR channel blocker, completely abolished spike initiation (*Figure 1b–d*). These results indicate that as in fine dendrites of hippocampal and neocortical pyramidal neurons (*Stuart and Spruston, 2015*; *Major et al., 2013*), NMDA spikes can be initiated in dendrites of pyramidal neurons in PCx. An ANOVA test for comparison of all four groups (control, TTX, Ca blockers and APV) showed statistical significance (p < 0.05). Post-hoc analysis (Dunnett test) comparing each of the three groups (TTX, Ca blockers and APV) to the control yielded a significant difference only for comparison of APV to the control (p < 0.05). Voltage gated sodium and calcium channels only minimally contributed to the slow component of the spike, while the majority of the current was carried by NMDAR channels (*Major et al., 2008*; *Schiller et al., 2000*).

## Initiation of NMDA spikes by activation of LOT inputs using focal synaptic stimulation

Having established with focal glutamate uncaging that pyramidal neurons in PCx are capable of producing NMDA spikes in their distal dendrites, we asked whether synaptic activation of LOT inputs can also trigger dendritic spikes. We directly activated LOT inputs using synaptic stimulation electrodes visually positioned within the LOT pathway in layer 1 (*Apicella et al., 2010*; *Franks and Isaacson, 2006*) (*Figure 2a*). The dendritic stimulus location was verified using calcium imaging which showed a low amplitude localized calcium transient in conjunction with a small subthreshold EPSP (*Figure 2b*). Gradually increasing the stimulus intensity led to a linearly increasing EPSP up to a threshold stimulation intensity, after which a dendritic spike was initiated (*Figure 2c–d*).

The average spike threshold evoked by LOT stimulation and recorded at the soma was 14.1 ± 1.6 mV and the dendritic spike amplitude and area under the voltage curve (hereafter 'area') measured at the soma was 27.1 ± 2.4 mV and 3817.8 ± 396.9 mV*ms respectively (mean ± SD; n = 48 cells; stimulation location 276.78 ± 55 μm from soma).

In line with the uncaging data, APV (50 μM) blocked the initiation of dendritic spikes by LOT inputs and linearized stimulus-response curves (*Figure 2c–e*). At just-suprathreshold stimulus intensity, APV decreased the response peak and area by 56.3 ± 3.8% and 88.5 ± 2.6% respectively (n = 17). Also similar to uncaging, in 22% of neurons we recorded a fast initial spike component (*Figure 2f*; amplitude 13.322 ± 0.516 mV, threshold 9.71 ± 0.79 mV, n = 8; 217.14 ± 29 μm from soma).

The average resting membrane potential was relatively hyperpolarized (−80.1 ± 1.43 mV), so that in many cases the NMDA spike remained subthreshold for somatic firing, however in some cases we could observe somatic firing as a result of the NMDA spike.

Similar NMDA spikes were observed with parasagittal slices and with high cloride (20 mM) intracellular recording conditions as described in *Bathellier et al. (2009)*. These spikes were robust, and completely blocked by the NMDAR blocker APV. The residual synaptic response was completely blocked with the AMPAR blocker CNQX (*Figure 2—figure supplement 1*).

Occasionally we observed spontaneous spike-like events, which resembled synaptically evoked spikes in shape, including a clear inflection at spike initiation (*Figure 2g–h*). This indicates that the basic circuitry of the piriform cortex can support the initiation of such spikes. The average

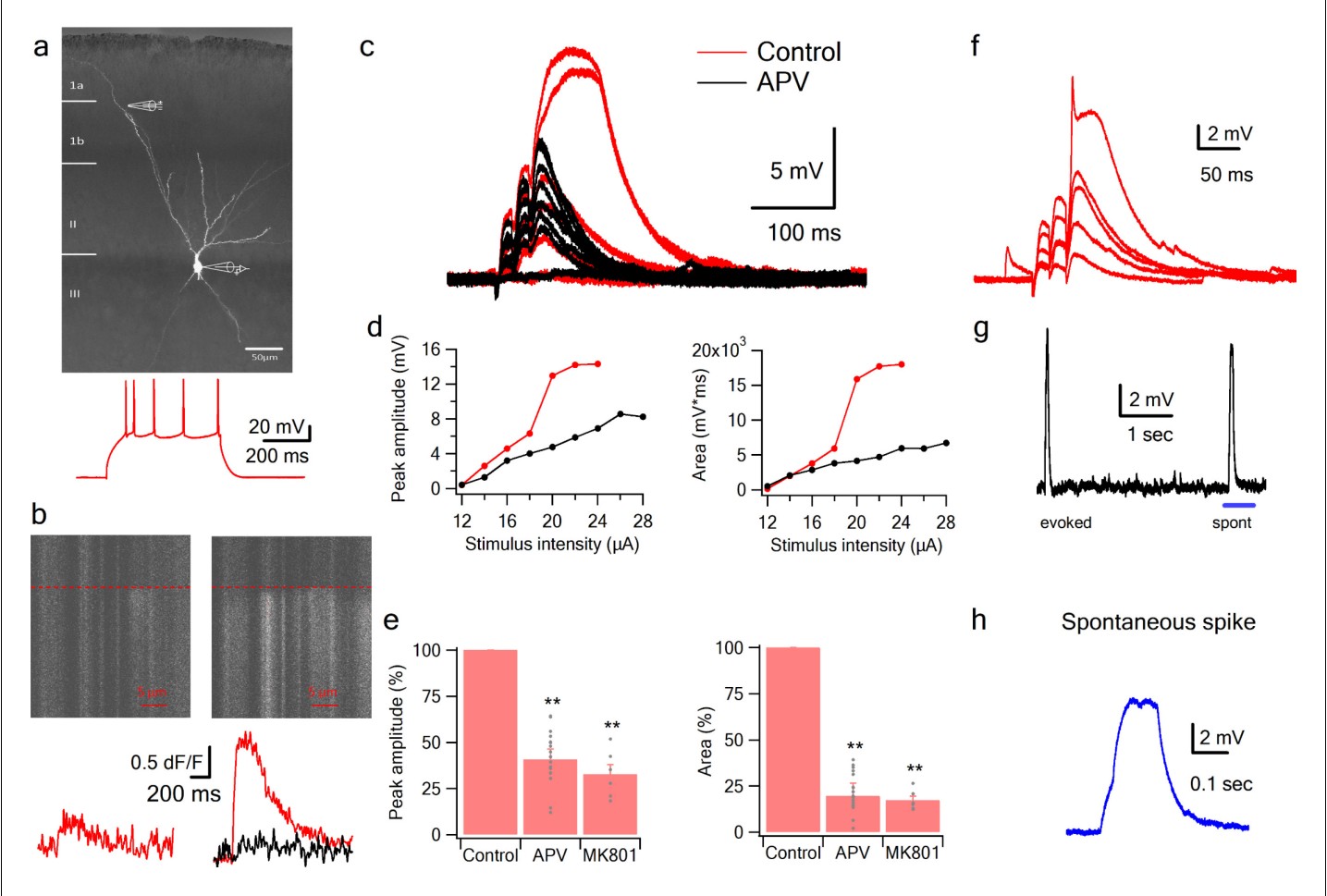

**Figure 2.** NMDA spikes evoked by LOT stimulation in PCx pyramidal neurons. (a) A pyramidal neuron from PCx was loaded with the calcium sensitive dye OGB-6F (200 µm) and CF633 (200 µm) via the patch recording electrode. A focal double-barreled synaptic stimulating theta electrode was placed distally within the LOT innervation zone (280 µm from soma). Bellow, is somatic action potential firing patter in response to somatic step current injection (500 ms). (b) Line scan crossing the dendrite close to the stimulating electrode, showing calcium transients for a subthreshold EPSP (left) and for a dendritic spike at the same site (right). Dashed line denotes the time when the stimulus was delivered. Bottom traces show calcium transients in Control (red) and after addition of APV (black) for a subthreshold EPSP (left) and a dendritic spike (right). (c) Voltage responses evoked by gradually increasing synaptic stimulation consisting of a burst of 3 pulses at 50 Hz. With gradually increasing stimulus intensity, an all-or-none response was evoked in control solution (red), which was blocked with the addition of the NMDAR blocker APV (50 µM, black). No biccuculine was added in this experiment. (d) Voltage response peak and area plotted as a function of stimulus intensity for the cell shown in A, showing a sigmoidal curve in the Control condition (red) and a linear curve with APV (black). (e) Summary plot mean (±SEM) for spike peak amplitude and area in Control (n = 48) and after APV application (n = 17) or intracellular MK801 (n = 6). The tip of the electrode was filled with 1 µL of control solution and back-filled with solution containing MK801(1 mM). (f) Example of a combined NMDA spike and fast spikelet probably representing a local sodium event. (g) Example of a spontaneous spike recorded in succession to the synaptically evoked spike denoted by the blue bar is shown with higher time resolution in (h) **p < 0.01 for comparison with control. Comparison between APV and MK801 did not reach statistical significance.

DOI: https://doi.org/10.7554/eLife.38446.003

The following figure supplement is available for figure 2:

**Figure supplement 1.** NMDA spikes in PCx pyramidal neurons: parasagittal slices.

DOI: https://doi.org/10.7554/eLife.38446.004

spontaneous spike amplitude and area were 17.78 ± 2.09 mV and 9941 ± 2640 mV*ms respectively (n = 22 spikes).

To further study the role of NMDARs in synaptically evoked spikes, we blocked NMDARs intracellularly. We and others have previously shown that intracellular MK801 can block NMDARs from the inside and thus can serve as a powerful tool to separate regenerative postsynaptic amplification

effects of NMDARs from recurrent network effects (*Behabadi et al., 2012*). Addition of MK801 to the patch pipette solution completely blocked dendritic spikes (*Figure 2e*). The spike amplitude and area at just-suprathreshold stimulation intensity was reduced to 32.7 ± 5.3% and 17.2 ± 2.2% respectively (n = 6 cells; 40 min after patch breakthrough).

Dendritic calcium imaging revealed that NMDA spikes were accompanied by large calcium transients around the activated dendritic site (*Figure 3*). Large calcium transients were seen only in the stimulated branches; calcium transients in unstimulated branches remained low (*Figure 2—figure supplement 1*). Using OGB-6F we observed a maximal calcium transient at the activated dendritic location which fell off steeply in both proximal and distal directions relative to the activated sites (*Figure 3b–e*). These data are consistent with NMDA spike-evoked calcium profiles seen in basal dendrites of layer five pyramidal neurons, and indicate the initiation of a local, non-actively propagated spike (*Major et al., 2008*; *Schiller et al., 2000*).

Together with the uncaging data, these results indicate that LOT inputs are capable of generating NMDA spikes in distal pyramidal neuron dendrites, and these spikes strongly amplify peak and time-averaged somatic EPSP responses compared to just-subthreshold (to NMDA spike) responses (218.4 ± 16% and 313.5 ± 34.1% for peak amplitude and area respectively).

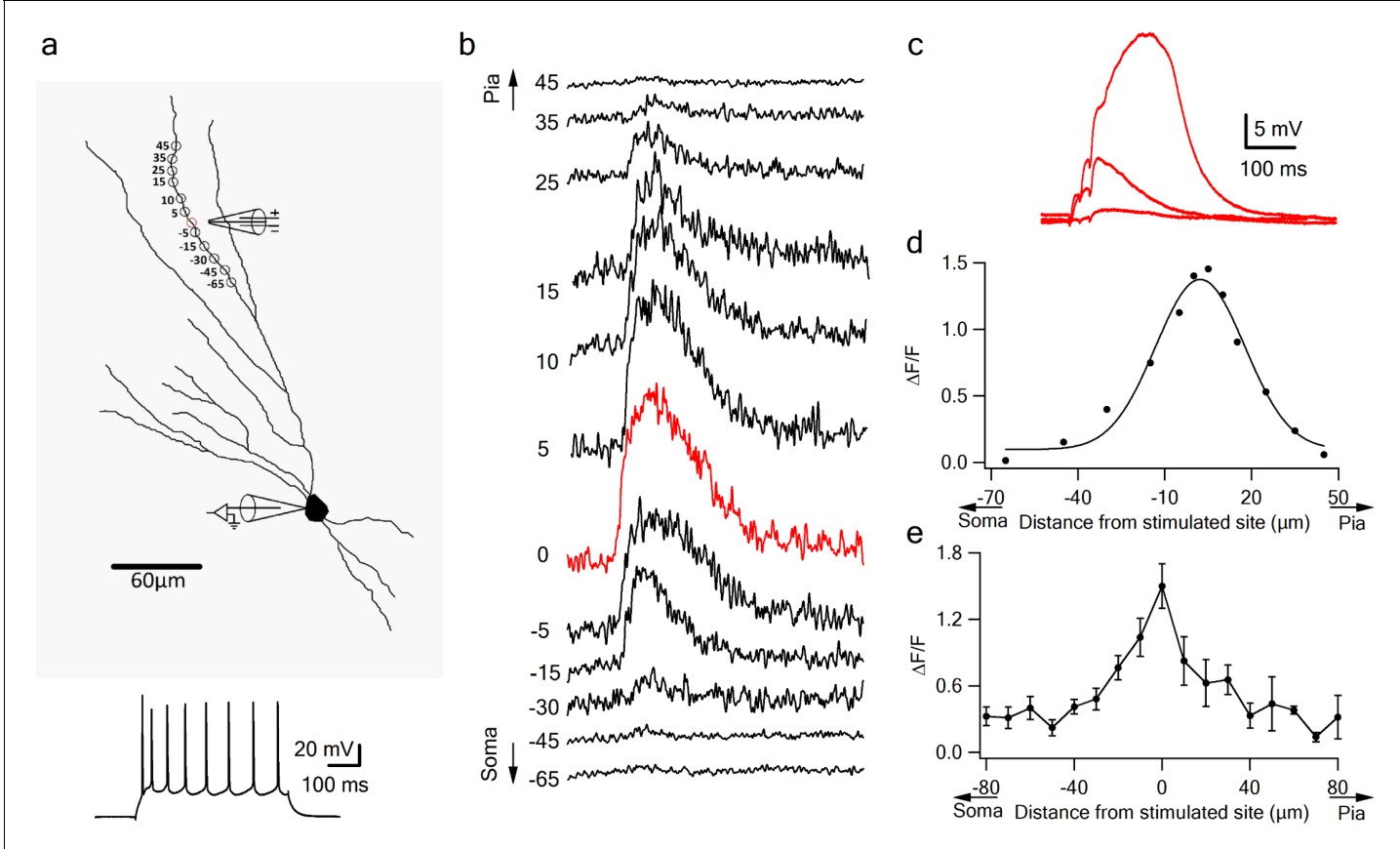

**Figure 3.** Local calcium transients evoked by dendritic NMDA spikes. (a) Fluorescence reconstruction of a layer IIB pyramidal neuron, showing stimulation electrode (288 µm from soma) and the sites of calcium imaging (red circle denotes the location of synaptic stimulation). Bellow, is somatic action potential firing patter in response to somatic step current injection (500 ms). (b) Calcium profile along the stimulated apical dendrite. Calcium transients are expressed as ΔF/F shown for different segments around the stimulated site (as illustrated in A). '0' denotes stimulation location, while distances (in µm) from stimulation site towards pia is indicated as +ve and towards soma as –ve. (c) Example of subthreshold EPSPs and an NMDA spike evoked at this recording site. d. Calcium profile (peak ΔF/F) fitted by a Gaussian curve plotted as a function of distance of spike location (for the experiment shown in a-b). (e) Summary plot of mean change in calcium transient (ΔF/F ± SEM) evoked by an NMDA spike, as a function of the distance from the center of a stimulated segment (0) averaged in 10 µm segments, from proximal (-ve values) to distal (+ve locations). All cells contained OGB-6F dye, perfused through the patch pipette (n = 5).

DOI: https://doi.org/10.7554/eLife.38446.005

## NMDA spikes can be generated throughout the apical tree, and by both LOT and IC inputs

In addition to bulb inputs conveyed by the LOT in layer 1a, pyramidal neurons in PCx receive a much larger number of inputs from IC axons in the deeper layers (*Haberly, 2001*; *Haberly and Price, 1978*; *Johnson et al., 2000*). Typically, an odor response in the piriform cortex is composed of feedforward LOT activity followed by recurrent IC activity, and it is thought that odor responses in PCx are strongly shaped by this recurrent input (*Davison and Ehlers, 2011*; *Poo and Isaacson, 2011*; *Franks et al., 2011*). To develop a more comprehensive picture of dendritic integration in pyramidal neurons of PCx, it is therefore important to determine whether IC inputs can also trigger NMDA spikes in pyramidal neuron dendrites.

Using glutamate uncaging, we first tested whether NMDA spikes could be initiated at progressively more proximal sites along the apical dendrites of PCx pyramidal neurons. We found that spikes could not only be generated throughout the LOT-recipient zone in layer 1a, but also in layers 1b and two where pyramidal neurons primarily receive intracortical inputs (*Figure 4*). In keeping with previous reports (*Major et al., 2008*), the amplitude of NMDA spikes recorded at the soma increased significantly (from $6.4 \pm 0.7$ mV to $25.9 \pm 3.1$ mV) as the uncaging site moved from distal ($318.6 \pm 7.9$ μm) to proximal ($99 \pm 14.6$ μm) dendritic locations (*Figure 4c*; p = 0.00014). Thus, the apical dendrites of PCx pyramidal neurons are capable of generating spikes throughout the layers that receive LOT and IC inputs.

To examine with greater specificity the relative contribution of LOT versus IC inputs to NMDA spike initiation at different distances from the soma, we used the GABA-B agonist baclofen (100 μM), which was previously shown to selectively silence intracortical inputs (*Apicella et al., 2010*; *Franks and Isaacson, 2005*; *Tang and Hasselmo, 1994*). Addition of baclofen did not change significantly the average resting membrane potential ($-75.51 \pm 2.77$ and $-75.77 \pm 2.74$ before and after addition of Baclofen. p > 0.5). When dendritic spikes were initiated at distal dendritic locations using synaptic stimulation ($254.85 \pm 14.95$ μm from soma), addition of baclofen only slightly altered spike amplitude (*Figure 4d–e*; spike amplitude was reduced by $10.4 \pm 7.3\%$, p = 0.02 and spike threshold increased by $10.1 \pm 12.6\%$, p = 0.004, n = 6). However, at mid and proximal apical dendritic regions, baclofen exerted substantial effects on spike initiation and voltage amplitude (*Figure 4f–h*). At mid dendritic locations ($197.3 \pm 12$ μm), local spikes were evidently triggered by a mixture of LOT and intracortical inputs, since upon baclofen application, response amplitude was reduced (peak voltage response was reduced by $42.0 \pm 13.3\%$, compared to control; n = 6). At more proximal locations ($161.6 \pm 9.6$ μm) spike initiation was almost completely dependent on intracortical inputs: when baclofen was present, we were unable to initiate local spikes at all, and the voltage response was significantly reduced (*Figure 4f–g*; peak response was reduced by $63.2 \pm 6.2\%$, compared to control; n = 6). At all dendritic locations, addition of APV (50 μm) completely abolished spike initiation (*Figure 4d–h*). However, at proximal locations APV did not significantly change the response amplitude recorded in the presence of baclofen (*Figure 4g–h*; reduction of $8 \pm 10.6\%$; p = 0.44; n = 5). Thus, mainly the regenerative part of the spike was suppressed by the intracortical blockade by baclofen, leaving only the underlying EPSP.

Taken together these results indicate that pyramidal neurons in PCx are capable of NMDA spike generation throughout their apical arbors; that both LOT and IC inputs can generate NMDA spikes within their respective layers; and that these synaptically evoked dendritic spikes show the commonly observed increase in amplitude as the initiation site moves closer to the soma (*Major et al., 2008*).

## Combination selectivity and compartmentalization of pyramidal neuron dendrites

Having a 'discontinuous' receptive field means an olfactory pyramidal neuron must respond selectively to multiple different combinations of LOT inputs (e.g. AB), but not to the individual inputs (A, B, or C), or to re-combinations of the same inputs (AC or BC). We tested a pyramidal neuron's capacity for responding selectively to multiple distinct LOT input combinations, in two stages.

First, we verified that the NMDA spike thresholding nonlinearity could provide a mechanism for enforcing combination selectivity *within* a dendrite. For example, a dendrite with a threshold of 2 could respond to a combination of Input one and Input two applied together, but not to the

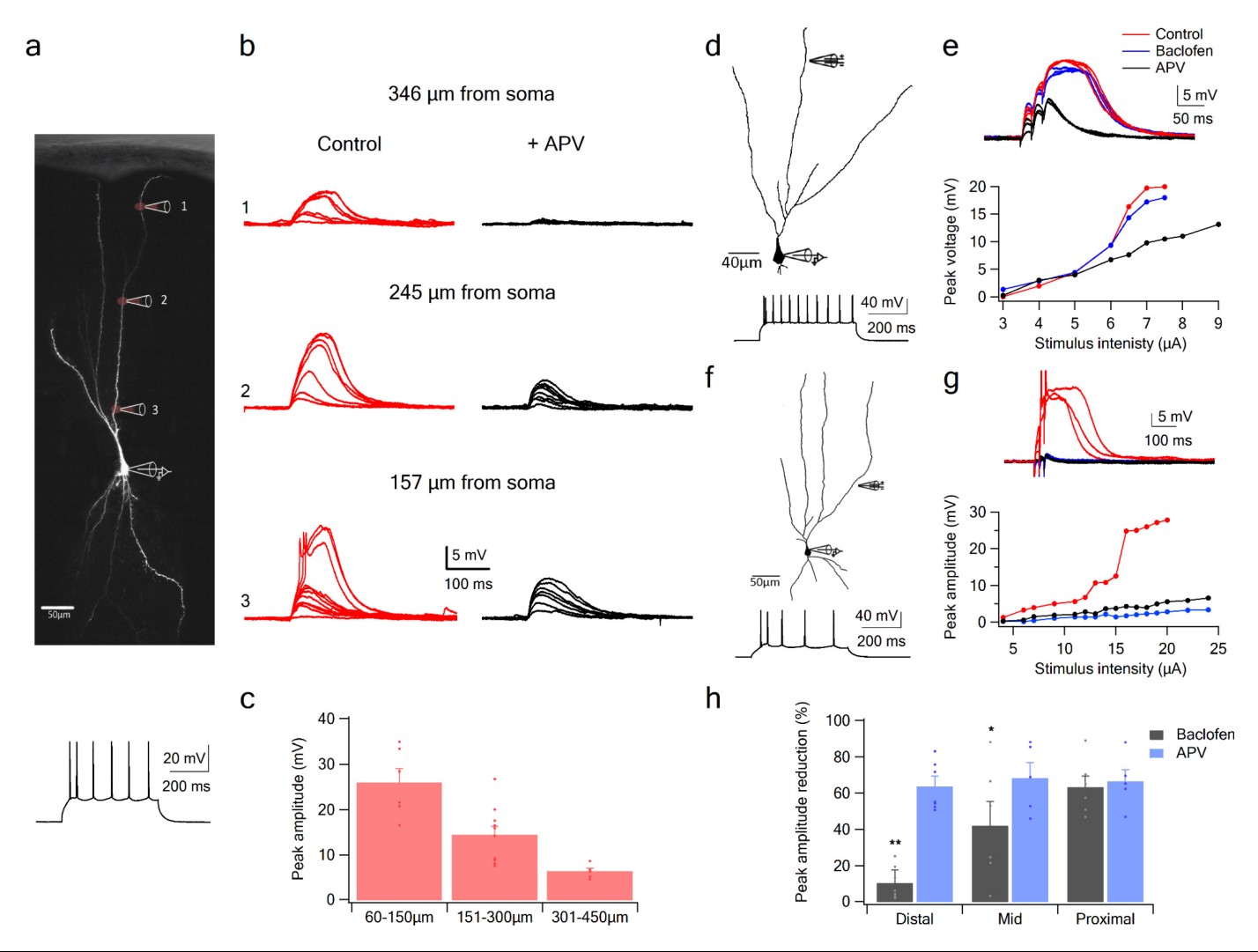

**Figure 4.** NMDA spike initiation with LOT and IC inputs. (a) Pyramidal neuron was loaded with the fluorescent dye CF-633 (200 μm) via the somatic patch recording electrode. Glutamate (MNI-glutamate) was uncaged at three sites as indicated by red circles (346 μM, 245 μM and 157 μm from soma). Bellow, is somatic action potential firing patter in response to somatic step current injection (500 ms). (b) Somatic voltage responses evoked by increasing laser intensity at the dendritic locations indicated in A, in Control (left, red) and with the blocker APV (right, black). (c) Summary plot of dendritic spike peak amplitudes as recorded at the soma, as a function of distance from the soma (n = 11). (d) Reconstruction of a pyramidal neuron showing a focal stimulation electrode at a distal LOT receiving zone (244 μm from soma). Bellow, is somatic action potential firing patter in response to somatic step current injection (500 ms). (e) NMDA spike in Control (red), after sequential addition of Baclofen (100 μM; blue) and APV (50 μM; black). Bottom, plot of peak voltage response as a function of stimulus intensity for Control (red), in the presence of baclofen (blue) and sequential addition of APV (black). (f) Reconstruction of a pyramidal neuron showing focal stimulating electrode at a proximal IC dendritic receiving zone (148 μm from soma). Bellow, is somatic action potential firing patter in response to somatic step current injection (500 ms). (g) NMDA spike in Control (red), after sequential addition of Baclofen (100 μM; blue) and APV (50 μM; black). Bottom, plot of peak voltage response as a function of stimulus intensity for Control (red), in the presence of baclofen (blue) and sequential addition of APV (black). (h) Summary plot of mean percent reduction in voltage amplitude measured at NMDA spike threshold in control conditions (mean % reduction peak voltage ± SEM) in the presence of baclofen and APV, for distal (n = 7), middle (n = 6) and proximal (n = 6) spike locations. *p < 0.05; **p < 0.01.

DOI: https://doi.org/10.7554/eLife.38446.006

individual components Input one or Input two applied separately. To test this, we used stimulating electrodes to activate two LOT inputs separately and in combination on a single dendrite (3 EPSPs at 50 Hz; 354.4 ± 17.47 μm from soma; average interelectrode distance of 35.69 ± 1 μm). Input one was activated over a full range of intensities, up through NMDA spike initiation. After generating Input 1's baseline input-output curve, the same stimulus sequence was repeated in the presence of

Input 2, which provided a constant 'bias' input (average EPSP bias amplitude at the soma was 3.81 ± 0.24 mV; n = 26). Coactivation of the two inputs resulted in a strong nonlinear interaction, where Input two significantly lowered the threshold for local spike generation by Input 1 (by 46.55 ± 1.8% for peak voltage; t-test p < 0.0001 and 52 degrees of freedom; F-score of 2.36; *Figure 5a–d*) without changing the response peak amplitude. The pronounced left shift of the input-output curve caused by the Input two bias input is a fundamentally nonlinear interaction resembling the function $sigmoid(I_1 + I_2, \theta)$, where $I_1$ and $I_2$ represent the magnitudes of the two inputs, and $\theta$ represents the threshold. Had the interaction been linear, the effect of Input two would have been to lift Input 1's entire input-ouput curve vertically by an amount equal to the bias voltage, with no change in threshold, as represented by the formula $sigmoid(I_1, \theta) + I_2$.

Given the form of the within-branch nonlinearity, we conclude that with an appropriate setting of the NMDA spike threshold, the distal dendrites of olfactory pyramidal neurons are well suited to enforce LOT combination-selectivity.

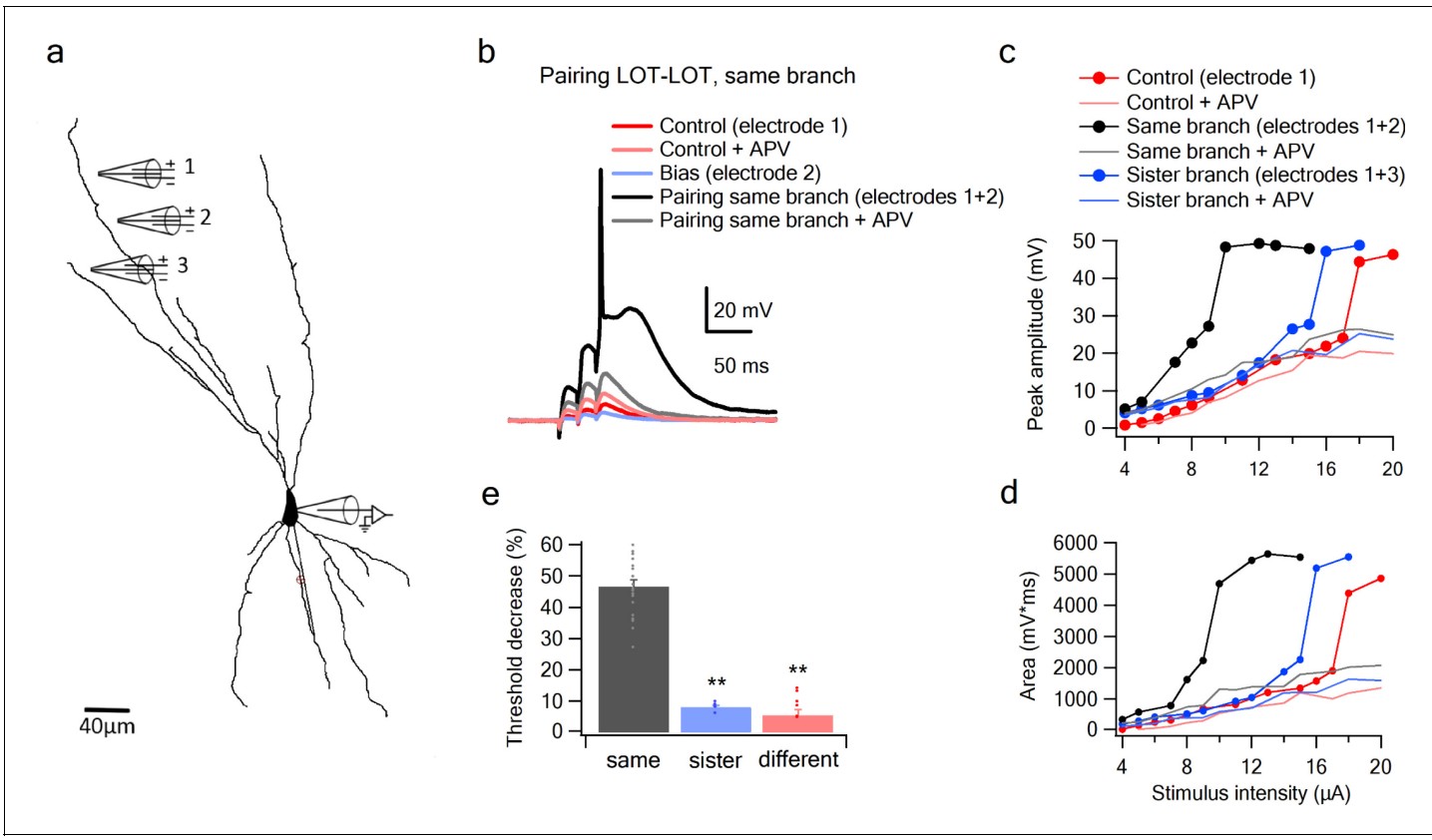

**Figure 5.** Summation of LOT inputs on same, sister and different dendritic branches. (a) Reconstruction of a layer IIB pyramidal neuron filled with the fluorescent dye CF-633 (200 µM) with stimulating electrodes positioned within an LOT-receiving zone in same dendritic branch (electrodes 1, 2) and sister branch (electrode 3). (b) Voltage responses to pairing LOT inputs in same branch (electrodes 1 + 2) in Control (black) and with APV (grey). Example responses to electrode one separately (red), electrode 2 (blue, bias voltage), and electrode one with APV (light red) are shown. c. Stimulus response curves (peak voltage) for paired LOT inputs within the same branch (electrodes 1 + 2; inter-electrode distance 38 µm) and pairing with a sister branch (electrode 1 + 3). d. Same as in C, for area of the responses. e. Summary plot of percent decrease of spike threshold in the paired condition relative to Control for LOT-LOT inputs in same branch (grey) sister branch (blue) and different branch (red). ** ANOVA test for comparison between the three groups (same, sister, different branches) showed statistical significance (p < 0.01). Post-hoc analysis (Tukey-Kramer test) comparing each group with every other group yielded significant differences between same branch and sister and different branches (p < 0.01), but no statistical significance between sister and different branches (p = 0.37).

DOI: https://doi.org/10.7554/eLife.38446.007

The following figure supplement is available for figure 5:

**Figure supplement 1.** Pairing activation of LOT inputs with glutamate uncaging at distal dendritic branches.

DOI: https://doi.org/10.7554/eLife.38446.008

The second stage question is whether a co-activation of two inputs of comparable magnitude to those used in the experiments above, but split *between* dendrites, are less effective at driving the cell than the within-branch combination. We defined sister branches as branching from same root dendrite coming of the soma and different branches as branching from two separate root branches. We found in favor of this hypothesis, that the nonlinear interaction between LOT inputs delivered to two different branches, even when those branches were 'sisters' (*Figure 5c–e*), was much weaker than the within-branch interaction, as evidenced by the much smaller change in spike threshold caused by the bias input (7.98 ± 0.75%, n = 5 threshold change for the sister-branch and 6.28 ± 1.88%, n = 6 threshold change for the different-branch, *Figure 5c–e*). An ANOVA test for comparison between the three groups (same, sister, different branches) showed statistical significance (p < 0.01). Post-hoc analysis (Tukey-Kramer test) comparing each group with every other group yielded significant differences between same branch and sister and different branches (p < 0.01), but no statistical significance between sister and different branches (p = 0.98).

To more closely examine the role of NMDA channels in the pairing outcomes, we repeated the pairing experiments in the presence of APV (*Figure 5c–d*). In these cases, the bias added a roughly constant value to the unpaired voltage response (Control + APV) all along the curve, confirming that without NMDA channels, pyramidal neuron dendrites revert to roughly linear summation.

To further examine summation of two LOT inputs at different dendritic locations, we replaced one of our stimulating electrodes with a glutamate uncaging spot (*Figure 5—figure supplement 1*). Similar to the previous two-electrode summation experiments, we observed a significant difference between same and different branch summation. The threshold was lowered by 52.2 ± 2.34% and 15.6 ± 4.2% when the synaptic bias was located on the same or different branches, respectively.

Together these results support a model of olfactory coding in PCx in which (1) NMDA spike generation in the distal apical dendrites of pyramidal neurons provides the superlinearity needed to enforce selectivity for specific combinations of LOT inputs, while the compartmentalization of voltage signals in the apical tree allows for different LOT combinations to be mapped onto different apical dendrites with relatively little crosstalk between them.

## Nonlinear summation of LOT and IC inputs

Given that odor responses in pyramidal neurons of the PCx are driven first by direct LOT activity and subsequently shaped by recurrent IC inputs (*Bekkers and Suzuki, 2013*; *Davison and Ehlers, 2011*; *Franks et al., 2011*; *Giessel and Datta, 2014*), it is critical to understand how LOT and IC inputs summate, whether linearly or nonlinearly, and if nonlinearly, with what type of nonlinear interaction.

Since LOT and IC inputs are segregated along the proximal-distal axis of a pyramidal neuron's apical dendritic tree, we used focal synaptic stimulation to examine the interaction between distal sites representing predominantly LOT inputs, and more proximal sites representing predominantly IC inputs. In these experiments, one electrode was positioned at a distal site and held fixed, while a second electrode, which again provided the bias input, was moved closer to the soma within the same branch (*Figure 6a*; average interelectrode distance of 138.46 ± 7.78 μm and bias voltage of 3.9 ± 0.27 mV; n = 13). We found that summation of LOT and IC inputs was very similar in form to the summation of two inputs confined to the LOT (*Figure 6a–e*): the IC bias input again led to a substantial threshold reduction for LOT inputs (35.44 ± 2.0%, n = 15), with no increase in response magnitude. Indeed, the threshold-lowering effect of the bias input was relatively constant within increasing separation of the two inputs (*Figure 6f*), indicating that over much of its length, a pyramidal neuron apical dendrite functions as a single, relatively location-insensitive integrative subunit. We also found that, just as for pairs of inputs confined to the LOT, the nonlinear interaction between LOT and IC inputs depended completely on NMDA regenerativity: blocking NMDARs linearized the input-output curves, destroying their sigmoidal form, and eliminating the basis for a superlinear within–branch interactions (*Figure 6*, grey curves).

To complete the picture, we examined the interaction between a distal LOT input on one branch and a proximal IC bias input either on a sister or a different branch (*Figure 7*). When the bias was provided by an IC input on a sister branch, the nonlinear interaction was evident, though significantly weaker compared to that seen with a same-branch IC bias (*Figure 7a–b,e*). Threshold reduction was 23.38 ± 2.34% (n = 8) for IC locations on sister branches and 18.4 ± 2.88% (n = 7) for IC bias on different branch (*Figure 7c–e*).

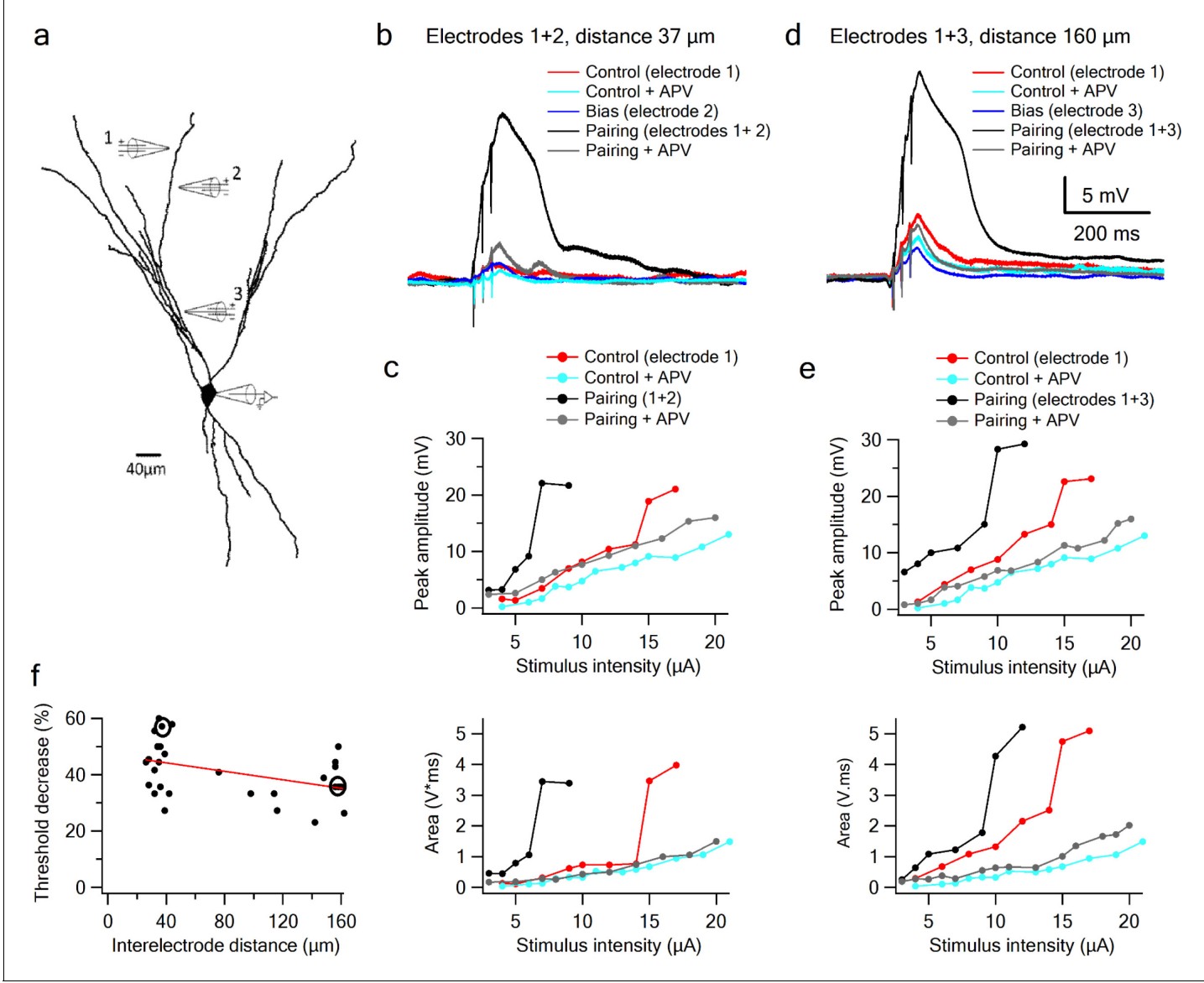

**Figure 6.** Supralinear summation of LOT and IC inputs on a single dendritic branch. (**a**) Reconstruction of a layer III pyramidal neuron filled with the fluorescent dye CF-633 (200 μM). Three stimulating electrodes where positioned in close proximity to same dendritic branch, with electrodes 1 and 2 within the LOT-receiving zone and electrode three within the IC-receiving zone. (**b**) Voltage responses for paired LOT inputs on the same branch (electrodes 1 and 2, inter-electrode distance 37 μm). Example responses are shown for electrode 1 (red), electrode 1 + APV (cyan), electrode 2 (blue, bias voltage), pairing activation of electrodes 1 + 2 (black), and pairing activation of electrodes 1 + 2 in the presence of APV. (**c**) Stimulus response curves for LOT pairing on the same branch (top voltage; bottom area). (**d**) Same as b, for pairing LOT and IC inputs on the same branch (electrodes 1 and 3, inter-electrode distance 160 μm). (**e**) Stimulus response curves for LOT and IC pairing on same branch (top voltage; bottom area). (**f**) Summary plot of percent decrease of spike threshold (paired condition relative to control) as a function of inter-electrode distance in same branch. Circles show the examples shown in the figure. Slope = −0.075 ± 0.03 mV/μm.

DOI: https://doi.org/10.7554/eLife.38446.009

An ANOVA test for comparison between the three groups (same, sister, different branches) showed statistical significance (p < 0.01). Post-hoc analysis (Tukey-Kramer test) comparing each group with every other group yielded significant differences between same branch and sister and different branches (p < 0.01), but no statistical significance between sister and different branches (p = 0.37).

In summary, summation between LOT-IC inputs within a single apical dendrite remains nonlinear, with only a slight weakening of the nonlinear interaction as the inputs are increasingly separated

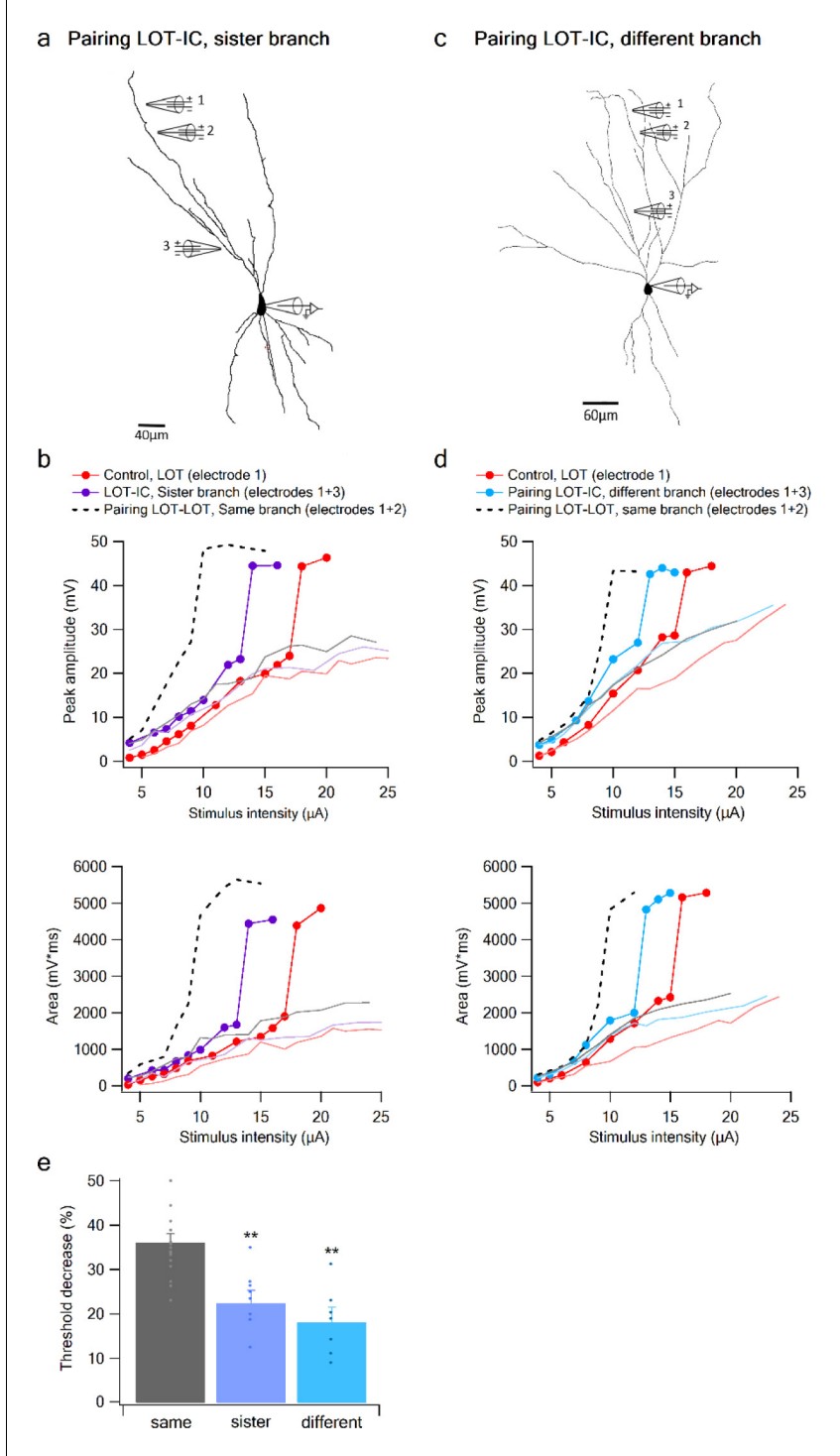

**Figure 7.** Summation of LOT and IC inputs on sister and different dendritic branches. (**a**) Reconstruction of a layer IIB pyramidal neuron (same as in *Figure 6*) filled with the fluorescent dye CF-633 (200 μM) with electrodes positioned within LOT regions of the same branch (electrodes 1,2), or at IC-receiving dendritic regions at sister branch (electrode 3). (**b**) Stimulus response curves for a single electrode (1, red), paired LOT-IC responses located on sister branches (1 + 3, purple), shown for peak amplitude (top) and area (bottom). Solid lines show same responses in the presence of APV. For comparison the paired LOT-LOT curve on the same branch (dotted black) is shown. (**c**) Reconstruction of a layer IIB pyramidal neuron filled with the fluorescent dye CF-633 (200 μM) with electrodes positioned within the LOT-receiving region of the same branch (electrodes 1,2), or at IC-receiving dendritic region at different branch (electrode 3). (**d**) Same as in B but for LOT and IC inputs activated on different

*Figure 7 continued on next page*

*Figure 7 continued*

branches. (**e**) Summary plot of percent decrease of spike threshold in paired condition relative to control for LOT-IC inputs to the same branch (grey), sister branch (blue), and different branch (teal). **p < 0.01.
DOI: https://doi.org/10.7554/eLife.38446.010

(compared to more closely spaced LOT-LOT pairs). This indicates that apical dendrites of pyramidal neurons in PCx function as relatively simple integrative subunits that apply a sigmoidal nonlinearity to their summed inputs with relatively little dependence on location. On the other hand, we observed a much weaker nonlinear interaction between inputs to different dendrites, indicating pyramidal neuron dendrites enjoy a significant degree of functional compartmentalization.

## Modeling

To cross check our experimental findings, we developed a compartmental model of a reconstructed PCx pyramidal neuron, and recorded its responses to stimulus configurations similar to those used in our experiments. We first established that the model cell can generate NMDA spikes in response to concentrated synaptic excitation at any location along a pyramidal neuron's apical dendrite, and that both the threshold for spike initiation and the spike amplitude measured at the soma increase as the site of spike initiation moves closer to the soma (*Figure 8—figure supplement 1*). We next replicated the same/sister/different branch pairing experiments shown in *Figures 5–7*. We found a close correspondence between the experimental and modeling results, wherein an LOT bias input activated on the same dendrite produced a much larger threshold-lowering effect than a bias input of the same size (measured at the soma) delivered to the LOT region of a sister or different branch (*Figure 8—figure supplement 2*). Thus, the model supports our experimental finding that nonlinear synaptic summation of LOT inputs to distal apical dendrites is strongly compartmentalized, with individual apical dendrites acting as well-separated integrative subunits.

We also found close correspondence to the experimental data for interactions between LOT and IC inputs on same, sister and different branches (*Figure 8—figure supplement 2*). The threshold lowering power of an IC bias input on the same branch was somewhat reduced compared to a bias input activated within the LOT itself, consistent with a mild distance-dependent attenuation of synaptic interactions on pyramidal neurons on same apical dendrites. In addition, the model replicated the experimental finding that IC bias inputs on sister and different branches had a significantly weaker effect than an IC bias on the same branch (compare *Figure 8—figure supplement 2g* to *Figure 8d*), though the degree of compartmentalization of IC inputs was, as in the experiments, and as expected from passive cable theory, less pronounced than the compartmentalization of LOT inputs. These nonlinear synaptic interactions persisted for non-synchronized inputs: our simulations revealed strong supralinear LOT and IC input summation when either of the inputs was temporally shifted by 20 ms, and in some cases more (*Figure 8—figure supplement 3*). Thus, the model supports our experimental findings that IC inputs interact nonlinearly with LOT inputs over extended spatio-temporal scales.

Finally, we used the model to verify that under in vivo-like conditions, the combined effects of NMDA spikes and dendritic compartmentalization can produce the 'discontinuous' receptive fields typical of pyramidal neurons in PCx. In particular, we predicted that combination selectivity would be observed for glomerular activation patterns that resulted in clustered excitation on apical dendrites, since this would tend to activate NMDA spikes and powerfully drive the cell, whereas input combinations that activated apical dendrites diffusely would fail to trigger NMDA spikes and therefore drive the pyramidal neuron only weakly.

To test this idea, we distributed LOT inputs in the distal apical tree in either a clustered or dispersed fashion, along with inhibitory inputs targeting both the distal apical dendrites (representing feedforward inhibition), as well as the perisomatic region (representing feedback inhibition) (*Figure 8a*). In addition to these stimulus-specific inputs, we randomly distributed an additional 100 excitatory and 20 inhibitory 'background' synapses over the entire apical dendritic tree to mimic the elevated background synaptic activity in-vivo. These 'background' synapses were driven by randomly generated temporal patterns and depolarized the cell by $11.2 \pm 2.3$ mV above the resting membrane potential (*Figure 8b*).

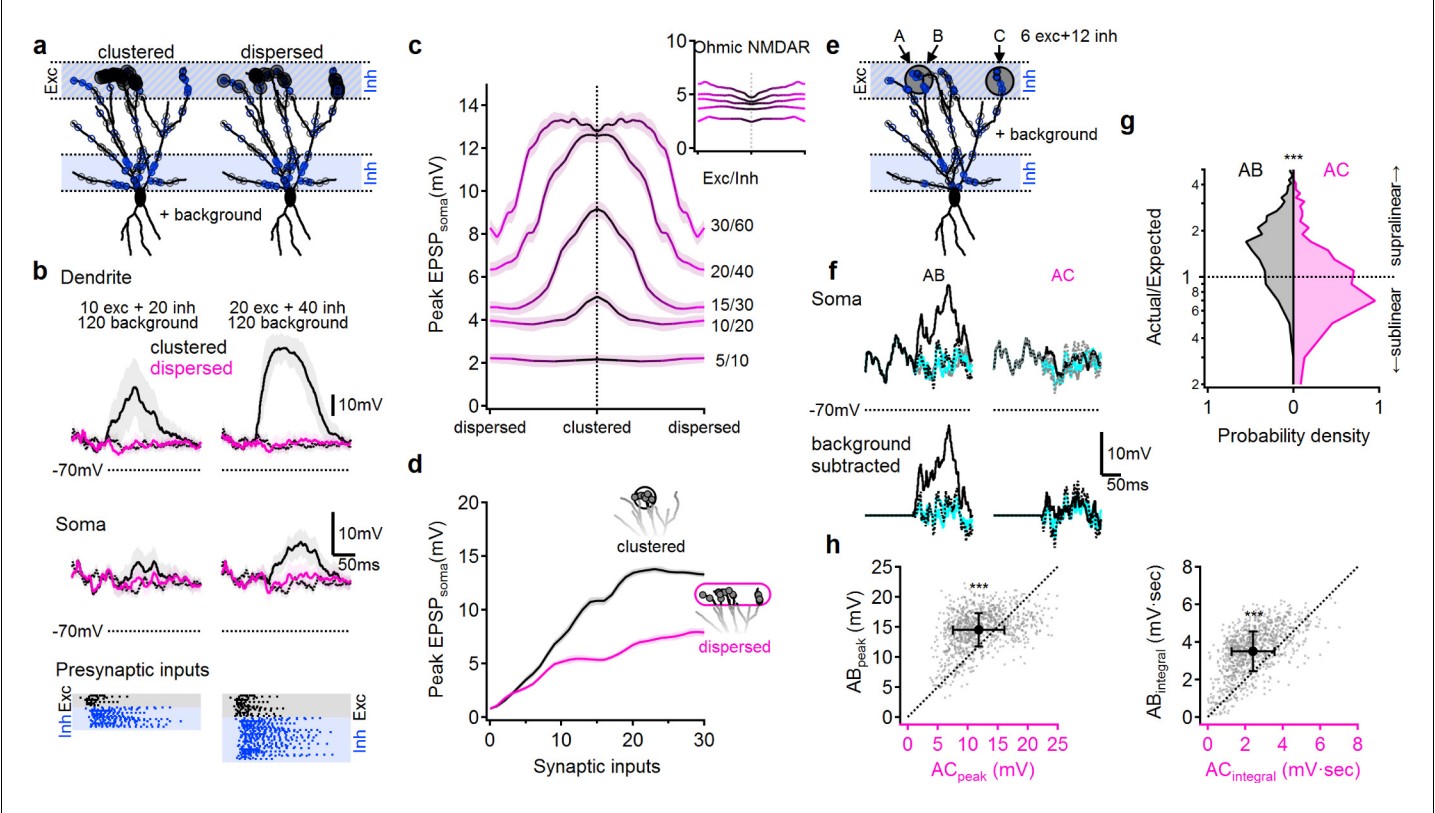

**Figure 8.** NMDA spikes can produce combination-sensitive dendritic receptive fields: modeling results. (a) Example distribution of glutamatergic inputs (grey circles) and GABAergic inputs (blue) on the reconstructed cell (for clarity, some dendrites are not shown). Olfactory information was mediated by 20 excitatory and 40 inhibitory inputs marked by closed circles. Open circles indicate additional 120 background synaptic inputs that were activated at random times. Left, clustered distribution, where stimulus-encoding excitatory inputs are concentrated on a single postsynaptic branch. Right, dispersed excitatory distribution over all distal dendrites. (b) Example postsynaptic responses to stimulation with 10/20 (left) and 20/40 (right) presynaptic excitatory/inhibitory inputs over a 120 input background. Top, voltages recorded from the stimulated dendrite. Middle, somatic EPSPs. Bottom, the temporal presynaptic activation pattern of signal–bearing inputs, simulated to mimic typical odor responses of mitral cells in-vivo. The presynaptic firing trains were identical between clustered and dispersed distributions. (c) Peak somatic EPSPs as a function of input clustering for different number of synaptic inputs. Blockage of NMDA spikes with Ohmic NMDA channels abolished the preference for clustered excitatory drive (n = 100 repetitions for each stimulation intensity) (d) The simulated peak EPSP amplitude recorded at the soma as a function of the number of presynaptic inputs. (e-h) Dendritic odorant selectivity with NMDA spikes. (e) Distribution of excitatory inputs (black) from three glomeruli (A-C, six inputs each; A and B target the same dendrite, while C innervated a different branch) and odor-unselective inhibitory inputs (24 synapses), on a background of 120 randomly activated background inputs. (f) Example somatic voltage profiles following activation of inputs from two glomeruli that converge on the same distal apical dendrite (AB, left) and glomerular input on two different dendrites (AC, right). Grey-background activity in the absence of LOT activation, magenta-somatic potential following activation of a single glomerulus, dotted black – expected linear summation, solid black – actual response to simultaneous pairing of two glomerular inputs. Top, original traces, bottom, with background activity subtracted. (g) pairing nonlinearity, quantified as the ratio between the area under the curve of the background subtracted actual response to the pairing vs. the area under the curve of the expected linear summation. Pairing nonlinearity of 1 represents a linear system, larger/smaller values represent supra/sub-linear responses respectively. n = 1000 simulation repeats, p < 10$^{-9}$ between AB and AC conditions, Kolmogorov-Smirnov test. (h) The peak (left) and the area under the curve (right) of the background-subtracted EPSPs in same (AB) vs. different (AC) dendritic pairing. Each data point represents a trail with similar spatial and temporal synaptic distribution of the background and stimuli inputs (save for the location of inputs B and C). n = 1000, p < 10$^{-9}$, paired t-test. Shaded areas-SEM.

DOI: https://doi.org/10.7554/eLife.38446.011

The following figure supplements are available for figure 8:

**Figure supplement 1.** NMDA spikes in PCx pyramidal neurons: modeling results.
DOI: https://doi.org/10.7554/eLife.38446.012
**Figure supplement 2.** Pairing LOT and IC inputs: modeling results.
DOI: https://doi.org/10.7554/eLife.38446.013
**Figure supplement 3.** A wide integration window for a simulated pairing of LOT and IC inputs:
DOI: https://doi.org/10.7554/eLife.38446.014

To further replicate in-vivo like conditions, LOT inputs were activated with in vivo-like firing patterns designed to mimic mitral cell responses to odors (*Figure 8b*, bottom (*Davison and Ehlers, 2011*; *Poo and Isaacson, 2009*; *Bolding and Franks, 2017*; *Tantirigama et al., 2017*);). The timing and kinetics of feedforward and feedback inhibitory inputs were likewise designed to mimic in-vivo like activation as described in the literature (*Figure 8b*) (*Poo and Isaacson, 2009*; *Sheridan et al., 2014*; *Suzuki and Bekkers, 2012*; *Sturgill and Isaacson, 2015*; *Large et al., 2016a*; *Large et al., 2016b*).

We compared clustered activation of LOT inputs (*Figure 8*, black) to the same number of synaptic inputs dispersed randomly over the entire LOT-receiving area of the dendritic tree (*Figure 8*, pink). As predicted, we found that that clustered LOT inputs reliably evoked NMDA spikes, powerfully amplifying postsynaptic signal compared to Ohmic (voltage-independent) NMDAR (*Figure 8c*). In contrast, dispersed LOT activation typically failed to overcome the local spike threshold on any dendrite, due both to the lack of concentrated excitation and the presence of inhibition, and therefore resulted in significantly smaller local dendritic and somatic voltage responses (*Figure 8b–d*, pink). The difference between clustered and dispersed synaptic distributions was evident over a large input range (*Figure 8c–d*), leading to preferential amplification of input combinations that target individual postsynaptic branches (*Figure 8c*). This effect was entirely dependent on regenerative NMDAR currents: stimulation with Ohmic NMDARs, eliminated the strong dependence of postsynaptic responses on the spatial distribution of synaptic inputs (*Figure 8c*, inset,).

Last, we tested whether the simulated neuron exhibited the fundamental capability needed to support a discontinuous receptive field: receiving equal inputs from multiple glomeruli, but responding only to specific combinations of them (*Figure 8e–h*). In keeping with the same-branch vs. different branch logic discussed above, we selected two dendrites as targets of stimulation from three glomeruli. Glomeruli A and B targeted the same terminal apical dendrite of the pyramidal cell, while glomerulus C innervated a different branch that originated from a distinct primary dendrite (*Figure 8e*). We then examined whether the cell would respond strongly to combination AB while ignoring combination AC, in line with our previous results and theoretical predictions. We found the AB combination produced strong supralinear responses (*Figure 8f* left, g, black), whereas the AC combination produced only a weak depolarization and mostly linear or sublinear summation (*Figure 8f*, right, g, pink). Thus, even in the presence of significant trial-to-trial variability introduced by random background activity and biologically-realistic presynaptic firing patterns, the model olfactory pyramidal neuron was able to use clustered glomerular activation as a basis for representing certain 'odor' combinations but not others (*Figure 8h*).

## Discussion

How pyramidal neurons in piriform cortex integrate their bulb inputs to generate olfactory percepts has been an unsettled question. To address this, we studied the integrative properties of pyramidal neuron dendrites in PCx using glutamate uncaging, focal synaptic stimulation, and compartmental models. Our primary aim was to determine whether local spike generation in the dendrites of PCx pyramidal neurons could serve the dual purposes of amplifying pyramidal neuron responses to LOT inputs impinging on their distal dendrites, as well as provide a nonlinear binding mechanism that could underlie a pyramidal neuron's selectivity for multiple distinct odorant combinations.

We found that pyramidal neurons in PCx can generate local spikes throughout their apical dendrites, and that a spike in a dendritic branch can powerfully depolarize the soma (i.e. producing up to 40–50 mV depolarizations depending on distance and cell size). Our results are consistent with two key properties of local spikes seen in thin (basal, apical oblique, and tuft) dendrites of pyramidal neurons in other cortical areas. First, the majority of dendritic spikes we observed in PCx pyramidal neurons are mediated primarily by NMDAR channels similar to thin dendrites in other pyramidal neurons (*Polsky et al., 2004*; *Larkum et al., 2009*; *Nevian et al., 2007*; *Lavzin et al., 2012*; *Antic et al., 2010*; *Schiller et al., 2000*; *Ariav et al., 2003*; *Major et al., 2013*). In addition, in a subset of cases we observed fast spikes reminiscent of sodium spikelets observed in thin tuft and basal dendrites of pyramidal neurons in other cortical areas (*Larkum et al., 2009*; *Losonczy and Magee, 2006*; *Major et al., 2008*; *Ariav et al., 2003*). Second, the NMDA spike amplitude (measured at the soma) increase progressively as the site of spike initiation moves closer to the soma (*Major et al., 2008*; *Behabadi et al., 2012*).

Beyond the properties of NMDA spikes per se, we found that the logic of synaptic integration in pyramidal neuron dendrites in PCx, particularly the pronounced difference between same-branch and between-branch summation, is also consistent with that seen in other types of pyramidal neurons (*Poo and Isaacson, 2009*; *Poirazi et al., 2003*; *Archie and Mel, 2000*). In particular, synaptic inputs to a PCx pyramidal neurons are processed via a 2-layer computation. First, LOT inputs are combined within individual dendrites as roughly a weighted sum (i.e. linearly) up to the local spike threshold (*Figure 2d*), so that an individual dendrite behaves comparably to a 'neuron' in a conventional artificial neural network. A telltale sign of this type of linear-nonlinear (LN) input-output function *within* a dendrite is the nearly pure left-shifting of a 'control' input's sigmoidal input-output curve caused by a constant 'bias' input activated on the same branch – an effect seen in both our experimental data and simulation results (*Figure 5*, and *Figure 8—figure supplement 1*). In the second layer of processing, the outflows from the separately thresholded dendritic 'subunits' are combined linearly at the soma as a prelude to output spike generation. A telltale sign of linear summation *between* dendrites is the uniform lifting of one branch's input-output curve (especially over its subthreshold range where saturation effects are minimal) by the constant somatic bias voltage generated by another dendrite. This is best seen in the peak amplitude curves in *Figure 7b and d*, and in the model input pairing figure (*Figure 8—figure supplement 1A* second sign of (relatively) independent functioning of different dendrites is the much smaller threshold shift in a dendrite's i/o curve seen when a bias input is applied to a different branch, where the least nonlinear crosstalk occurs between LOT-receiving zones in 'cousin' branches, and only slightly more between directly adjoining 'sister' branches (*Figures 5e* and *7e*, and *Figure 8—figure supplement 1g*; see also *Behabadi and Mel, 2014*).

The 2-layer architecture of a pyramidal neuron in PCx allows it to respond selectively to specific high-order odorant combinations – those whose LOT activity patterns deliver concentrated (suprathreshold) excitation to at least one apical dendrite – without responding to the vast majority of LOT patterns that produce more diffuse, and therefore subthreshold, excitation to multiple branches within the dendritic arbor. This ability to respond to multiple distinct high-order combinations, without responding to re-combinations of the same odor components, may account for this cell type's hallmark physiological property, namely its 'discontinuous' receptive field (*Stettler and Axel, 2009*; *Rennaker et al., 2007*; *Kadohisa and Wilson, 2006*; *Poo and Isaacson, 2009*). This same type of scenario, in which a neuron computes a disjunction over a set of nonlinear 'features' mapped onto different dendrites has been previously proposed to underlie the pooling of multiple simple cell-like subunits within a complex cell's receptive field in V1 (*Archie and Mel, 2000*; *Mel et al., 1998*); the pooling of higher-order feature conjunctions in a memory circuit (*Poirazi et al., 2003*; *Wu and Mel, 2009*; *Legenstein and Maass, 2011*; *Morita, 2008*), and as a means to multiplex one of several neural pathways through to a cell's output, as might occur in the context of a decision task (*Yang et al., 2016*).

## Mismatch to previous results

As previously discussed, our results showing that pyramidal neurons in PCx have a 2-layer summation logic arising from (1) NMDA regenerativity and (2) a compartmentalized dendritic tree, are inconsistent with a previous study in olfactory cortex which reported little sign of regenerative NMDA or sodium currents in pyramidal neuron dendrites, and concluded that pyramidal neurons in PCx integrate their inputs essentially linearly (*Bathellier et al., 2009*). The reasons for the discrepancy between the Bathellier et al. study and ours are unknown, and cannot stem from differences in slicing and solution composition conditions (see *Figure 2—figure supplement 1S*). Rather, may result from a more distributed axonal recruitment compared to our stimulation conditions as evident by the large AMPA responses in their recordings (see their *Figure 6*).

## Transformation of the odor code from olfactory bulb to piriform cortex

Anatomical data indicate that (1) axons originating in the olfactory bulb and traveling along the LOT terminate broadly throughout the piriform cortex, targeting a dispersed population of pyramidal neurons, and (2) single pyramidal neurons receive inputs from multiple broadly distributed olfactory glomeruli (*Miyamichi et al., 2011*; *Nagayama, 2010*; *Sosulski et al., 2011*). Lacking any evidence for a patterning of these connections, the anatomical projection from the olfactory bulb to the

cortex is generally assumed to be random. In keeping with this assumption, physiological data show that different odors activate a unique set of neurons widely distributed across the PCx, and that individual olfactory pyramidal neurons respond to a small, unpredictable subset of odors (*Stettler and Axel, 2009*; *Roland et al., 2017*). Thus, whereas the bulb forms a molecular-based code wherein M/T cells in a given glomerulus respond to any odor containing that glomerulus' associated molecule, pyramidal neurons in PCx fire only to high-order glomerular combinations, but respond to multiple combinations that appear to have little to no chemical overlap with each other.

This transformation from molecule-specific responses in olfactory glomeruli to multi-combination selectivity in pyramidal neurons in PCx requires a nonlinear transformation, and indeed several in-vivo and in-vitro studies have indicated that single pyramidal neurons in PCx do integrate odor information nonlinearly (*Davison and Ehlers, 2011*; *Stettler and Axel, 2009*; *Apicella et al., 2010*; *Poo and Isaacson, 2011*). In particular, activation of a single glomerulus or LOT fiber generates only a small subthreshold depolarization in most connected PCx pyramidal neurons. On the other hand, when multiple glomeruli or multiple LOT inputs are activated simultaneously, or combinations of odorants are presented to an animal, PCx pyramidal neurons produce strong supralinear responses (*Davison and Ehlers, 2011*; *Stettler and Axel, 2009*; *Kadohisa and Wilson, 2006*; *Apicella et al., 2010*; *Wilson, 2003*).

What mechanism underlies the supralinear integration of convergent LOT inputs onto single PCx pyramidal neurons? Apicella et al (*Apicella et al., 2010*). showed that the supralinear summation of LOT inputs onto pyramidal neurons, required neither cooperative interactions in the bulb, nor participation of recurrent IC inputs in PCx. However, the nonlinearity described by Apicella is consistent with either dendritic amplification mechanisms or axonal thresholding effect. Indeed, another potential source of supralinearity would be the cell's output spiking mechanism: a high threshold for somatic action potential generation could in principle be used to limit a pyramidal neuron's responses to only those stimuli that activate N (or more) connected LOT axons. However, without dendritic subunitization, the cell should respond to *any* combination of N LOT inputs, destroying the combination selectivity needed to account for a PCx pyramidal neuron's discontinuous RF. In contrast, our results support the idea that the dendritic thresholding nonlinearity provided primarily by regenerative NMDA currents can mediate the supralinear integration of LOT inputs observed previously both in-vitro and in vivo.

## The softer compartmentalization of IC inputs, and its implications

In our exploration of the nonlinear interactions between driver inputs within the LOT and bias inputs delivered either within the LOT or at the IC-receiving regions of the apical tree, we found that the threshold-lowering effects of IC inputs were less well compartmentalized. The effect can be traced to passive cable theory: IC inputs are closer to the branch points where sister and cousin dendrites connect to each other, so that their effects are felt more widely. In quantitative terms, beginning with a 'control' input-output curve generated by an LOT input, we found that the threshold-lowering power of a second LOT input on the same branch was roughly 10 times that of an LOT input delivered to a different branch. In contrast to this strong compartmentalization, the threshold-lowering effect of an IC bias input on the same branch is only twice that of an IC bias delivered to a different branch (*Figure 7e*). The observation that IC inputs modulate more globally comes with a caveat, however: it was previously shown that the degree of nonlinear crosstalk between dendritic branches tends to be overestimated in subthreshold summation experiments, compared to a cell operating in the firing regime which enhances subunit independence (*Behabadi and Mel, 2014*). This is because the somatic spike-generating mechanism acts as a sort of time-averaged 'voltage clamp' (*Holt and Koch, 1997*) that suppresses subthreshold voltage communication between dendrites (*Behabadi and Mel, 2014*). In light of this effect, it remains to be determined whether the softer compartmentalization of IC inputs seen in both our experiments and simulations will persist to the same degree under normal operating conditions in the olfactory cortex.

Both the existence of nonlinear interactions between feedforward LOT and recurrent IC inputs to PCx pyramidal neurons, and the (unknown) degree to which IC inputs act locally (i.e., have modulatory effects confined to a single dendrite) vs. globally (affecting some or all dendrites) in vivo, suggest there remains much to learn about the functioning of the recurrent odor recognition network in piriform cortex. Besides carrying feedback from other pyramidal neurons in the area, IC inputs provide contextual information from higher-order cortical regions including the entorhinal cortex,

orbitofrontal cortex and amygdala, potentially allowing the assignment of cognitive and emotional value to odors (*Gottfried, 2010*; *Johnson et al., 2000*). The nonlinear interaction of LOT and IC inputs mediated by NMDA regenerativity could provide a biophysical mechanism for binding odor with contextual information in piriform cortex. If so, the dendritic subunitization of PCx pyramidal neurons, and the possibility of some locality of IC modulation within a neuron, could allow contextual information to be bound to certain odorant combinations represented by a neuron and not others.

### Possible role of NMDA spikes in dendrite-specific plasticity induction

Backpropagating action potentials in apical dendrites of PCx pyramidal neurons attenuate significantly as they propagate (*Bathellier et al., 2009*; *Johenning et al., 2009*), making it less likely that bAPs contribute to spike timing dependent plasticity of distal LOT synapses. In contrast, given that NMDA currents can produce large localized calcium transients in apical dendrites, confined to within ±20 µm of the activated site, such spikes could serve as local induction signals for plasticity of LOT synapses. In accordance with this notion, it was recently shown that NMDA spikes contribute to long-term potentiation (LTP) in the dendrites of CA3 neurons (*Brandalise and Gerber, 2014*). In the same way, a group of LOT synapses that fire together on the same dendrite in PCx could trigger a local plasticity event that induces LTP of the activated synapses. When the same odor is re-encountered at a later time, and re-activates the group of now potentiated synapses, an even more powerful NMDA-dependent response might be generated, signaling odor recognition (for a discussion of related ideas see (*Wu and Mel, 2009*; *Weber et al., 2016*)). Further work will be needed to determine the ways and conditions in which synaptic plasticity contributes to the learning-related functions of the olfactory cortex.

## Materials and methods

### Electrophysiology and calcium imaging

Coronal brain slices 300 µm thick from a 28–40 day old Wistar rats (male and femal) were prepared from the anterior part of the piriform cortex in an ice-cold artificial cerebro-spinal fluid (ACSF) solution saturated with 95% oxygen and 5% $CO_2$. The ACSF solution contained (in mM) 125 NaCl, 25 NaNCO$_3$, 25 Glucose, 3 KCl, 1.25 NaH2PO$_4$, 2 CaCl$_2$, 1 MgCl$_2$ PH 7.4. The slices were incubated for 30 min at 37°C and kept at room temperature afterwards. During experiments, cells were visualized with a confocal scanning microscope equipped with infrared illumination and Dot gradient contrast video microscopy. Whole cell patch clamp recordings were performed using an Axon amplifier (Multi clamp). For patching, glass electrodes (6–8 MΩ) were made from thick-walled (0.25 mm) borosilicate glass capillaries on a Flaming/Brown micropipette puller (P-97; Sutter Instrument). Intracellular pippet solution contained (in mM) 135 K+-gluconate, 4 KCl, 4 Mg-ATP, 10 Na2-phosphocreatine, 0.3 Na-GTP, 10 HEPES, 0.2 OGB-6F, 0.2 CF 633, and biocytin (0.2%) pH7.2. In some experiments (*Figure 2—figure supplement 1*) the intracellular pippet solution contained (in mM) 115 K+-gluconate, 20 KCl, 2 Mg-ATP, 2 Na-ATP, 10 Na2-phosphocreatine, 0.3 Na-GTP, 10 HEPES, 0.2 OGB-6F, 0.2 CF 633, and biocytin (0.2%) pH7.2.

Fluorescence confocal microscopy (Olympus FV1000) was performed on an upright BX61WI Olympus microscope equipped with a 60X (Olympus 0.9 NA) water objective. Neurons were filled with the calcium-sensitive dye OGB-6F (200 µM; Invitrogen) and CF 633 (200 µM; Biotium) to visualize the apical dendritic tree. Calcium transients were recorded in line-scan mode at 500 Hz.

All experiments were performed at 34° C.

All animal procedures were in accordance with guidelines established by the NIH on the care and use of animals in research and were confirmed by the Technion Institutional Animal Care and Use Committee.

### Focal stimulation

Focal synaptic stimulation, at apical dendrites of PCx pyramidal neurons was performed via a theta-glass (borosilicate; Hilgenberg) pipettes located in close proximity to the selected dendritic segment guided by the fluorescent image of the dendrite and the DIC image of the slice. The theta-stimulating electrodes were filled with CF-633 (Biotium; 0.2 mM). Current was delivered through the electrode (short burst of 3 pulses at 50 Hz), via stimulus isolator (ISO-Flex; AMPI). The efficacy and

location of the stimulation was verified by simultaneous calcium imaging evoked by small EPSPs and their localization to a small segment of the stimulated dendrite.

### Glutamate uncaging

MNI-glutamate (Tocris, Bristol, UK) was delivered locally near by a dendritic region of interest using pressure ejection (5–10 mbar) from an electrode (2 µm in diameter) containing 5–10 mM caged glutamate. Electrodes were positioned 20–30 µm from the dendrite of interest and caged glutamate was photolyzed by a 1 ms laser pulse (375 nm Excelsior, Spectra Physics) using the point scan mode (Olympus FV1000). Simultanouse calcium imaging was performed from the uncaged dendritic region.

### Drug application

In all pairing experiments, gamma-aminobutyric acid (GABAA) (1 µM bicuculline; Sigma) was added to the ACSF perfusion solution. In all uncaging experiments and in part of the synaptically evoked NMDA spikes experiments bicuculline was omitted, see for example *Figure 2*. In some experiments as indicated in the text, a cocktail of calcium channels blockers was added to the ACSF solution containing w-agatoxin 0.5 µM (P/Q type calcium channel blocker), conotoxin-GVIA 5 µM (N type calcium channel blocker), SNX 482 200 nM (R type calcium channel blocker) and nifedipine 10 µM (L type calcium channel blocker). Sodium channel blocker TTX 1 µm was applied to the ACSF solution. NMDA-R antagonist APV (50 µM, Tocris Bioscience) was added to the ACSF solution. In some experiments the NMDAR channel blocker MK801 (1 mM) was addedd to the intracellular solution, and the tip of the electrode was backfilled with control intracellular pipette solution.

### Statistcal procedure

The sample size was chosen based on standards used in the field using similar experimental paradigms. Importantly most of our experiments involve examining a variable on the same neuron and thus the sources of variability are smaller in these type of experiments.

Analysis was done with IgorPro (5.01; WaveMetrics), Exel and Clampfit (Molecular Devices) and Prism 7 (Graphpad) commercial softwares. Data are presented as mean ± SEM. For testing statistical significance we used two-tailed paired Student's t test. No statistical methods were used to predetermine sample sizes. Our sample sizes are similar to those reported in previous publications. Neurons were excluded in case the viability of the cell was compromised as monitored by resting membrane potential and shape and amplitude of action potentials evoked by current injection. Average resting membrane potential was −80.1 ± 1.43 mV. In addition, we excluded neurons in which the quality of the recordings deteriorated as measured by the access resistance.

### Modeling

The simulations were conducted in a compartmental model using the NEURON 7.4 simulation platform. Three pyramidal cells were reconstructed from z-stacks of fluorescently labeled neurons using Simple Neurite Tracer (ImageJ, *Longair et al., 2011*). The cells were subdivided into 493–544 compartments, with a maximum length of 19 µm. The soma area was 829 µm (*Isaacson, 2010*), the total dendritic length was 2238–2516 µm (*Mombaerts et al., 1996*) (*Buonviso et al., 1991a*). The resting membrane potential was −70 mV; the membrane resistance was 25,000 Ω·cm (*Mombaerts et al., 1996*); the axial resistance was 100 Ω·cm and the membrane capacitance was set to 1 µF/µm (*Buonviso et al., 1991b*). The simulations that included sodium and potassium voltage-gated currents used the Hodgkin-Huxley kinetics formalism. Specifically, fast sodium channels (reversal potential = 50 mV, gNa = 1000 mS/cm$^2$), and delayed rectifier and slow non-inactivating potassium channels (reversal potential = −87 mV, gKdr = 500 mS/cm$^2$, gKs = 20 mS/cm$^2$) were used to allow for spike generation and adaptation, respectively (*Lavzin et al., 2012*).

To study the synaptic integration in PCx pyramidal neurons, we modeled physiologically realistic patterns of synaptic bombardment. We distributed the excitatory synapses from LOT presynaptic cells in the PCx pyramidal neurons according to known anatomical and physiological properties (*Bekkers and Suzuki, 2013*; *Davison and Ehlers, 2011*; *Franks and Isaacson, 2006*; *Suzuki and Bekkers, 2012*). The firing rates and the number of spikes for individual LOT presynaptic cells were drawn from random distributions that matched the known in-vivo firing properties of Mitral/PCx

pyramidal cells (*Davison and Ehlers, 2011*; *Poo and Isaacson, 2009*; *Bolding and Franks, 2017*; *Tantirigama et al., 2017*). The stimulation intensity of a single presynaptic cell was set to produce an EPSC of 30 ± 19 pA, corresponding to a 1.4 ± 0.82 mV somatic EPSP (*Figure 8—figure supplement 1b*; *Suzuki and Bekkers, 2011*). Inhibitory inputs were randomly placed either in the LOT recipient band, even with the LOT excitation, or in the proximal apical dendritic region within 100 µm of the soma (*Large et al., 2016b*). For a clustered synaptic distribution, all excitatory inputs were placed on a single distal dendrite. In the dispersed distribution, excitatory inputs were allowed to target any LOT-recipient branch. To model intermediate clustering levels, we divided the excitatory input into two pools, one clustered and the second dispersed, and changed the proportion between the number of inputs in each pool. Background activity was mediated by a separate synaptic population consisting of 100 excitatory (AMPA-only) and 20 inhibitory inputs distributed randomly over the apical tree. Background activation was random and unique between different trials with a mean firing rate of 10 Hz. The net result of the background activity was a ~ 11 mV depolarization of the postsynaptic resting potential. Presynaptic neurons were represented by NetStim processes that generated temporal triggers for synaptic activation. Each presynaptic cell gave rise to a single synapse on the modeled cell. Synaptic inputs were driven by a unique spike train for each presynaptic cell, which was generated by setting the 'noise' parameter of the NetStim process to 0.5. Excitatory spike trains began at simulation time of 100 ± 6 ms, and inhibitory inputs followed 10 ms later (110 ± 6 ms) (*Poo and Isaacson, 2009*; *Suzuki and Bekkers, 2012*). The ISI and the number of presynaptic action potentials in the excitatory/inhibitory presynaptic populations were described by normal distributions (mean ± SD) of 8 ± 5 ms and 5 ± 3/10 ± 3 respectively (*Davison and Ehlers, 2011*; *Poo and Isaacson, 2009*; *Sturgill and Isaacson, 2015*).

Excitatory postsynaptic synapses contained AMPA-Rs and NMDA-Rs. Inhibition was mediated by GABA-A synaptic currents. GABA-A currents had an instantaneous rise time, a decay time of 7 ms, unitary conductance of 2 nS and reversal potential of −70 mV. All excitatory inputs reversed at 0 mV. AMPA-R currents had an instantaneous rise time and a decay time of 1.5 ms. The average unitary AMPA-R conductance was 1 nS (*Bathellier et al., 2009*). NMDA-R currents had a rise time of 2 ms and a decay time of 80 ms, and the average NMDA-R conductance was 2 nS. The NMDA-R conductance voltage dependence was modeled as follows: $gNMDA = 1/(1 + 0.25 \cdot exp(-0.08 \cdot V_m))$ where $V_m$ is the local membrane potential. In some simulations we canceled out the voltage dependence of the NMDA-R current by setting the $V_m$ to −70 mV for the whole duration of the simulation (*Lavzin et al., 2012*). Presynaptic vesicular release was explicitly modeled; each synapse was assumed to contain five vesicles, each with an independent release probability (Pr) of 0.1. The presynaptic pool was replenished with a rate of 100 sec[−168]).

## Acknowledgements

We thank Y Schiller for helpful discussions throughout the project and helpful comments on the manuscript. We thank Irena Reiter for excellent technical assistance and processing the biocytin-filled neurons. This study was supported by Israeli Science Foundation (JS), the Rappaport Foundation (JS), the Adelis Fund for Brain Research at Technion and the Allen and Jewel Prince Center (JS).

## Additional information

### Funding

| Funder | Grant reference number | Author |
| --- | --- | --- |
| Israel Science Foundation | | Jackie Schiller |
| Rappaport Foundation | | Jackie Schiller |
| Allen and Jewel Prince Center | | Jackie Schiller |
| Adelis Foundation | Brain Research Award | Jackie Schiller |

The funders had no role in study design, data collection and interpretation, or the decision to submit the work for publication.

## Author contributions
Amit Kumar, Data curation, Formal analysis; Oded Schiff, Data curation, Software, Formal analysis; Edi Barkai, Resources, Writing—original draft, Project administration; Bartlett W Mel, Conceptualization, Methodology, Writing—original draft; Alon Poleg-Polsky, Software, Formal analysis, Methodology, Writing—original draft; Jackie Schiller, Conceptualization, Formal analysis, Supervision, Funding acquisition, Writing—original draft, Project administration

## Author ORCIDs
Amit Kumar (iD) http://orcid.org/0000-0003-0674-3641
Edi Barkai (iD) http://orcid.org/0000-0002-7325-4269
Alon Poleg-Polsky (iD) https://orcid.org/0000-0003-1327-5129
Jackie Schiller (iD) http://orcid.org/0000-0001-9182-7166

## Ethics
Animal experimentation: All animal procedures were in accordance with guidelines established by the NIH on the care and use of animals in research and were confirmed by the Technion Institutional Animal Care and Use Committee (IL-012-01-18, valid until 10/4/2022).

## Decision letter and Author response
Decision letter https://doi.org/10.7554/eLife.38446.017
Author response https://doi.org/10.7554/eLife.38446.018

# Additional files

## Supplementary files
• Transparent reporting form
DOI: https://doi.org/10.7554/eLife.38446.015

## Data availability
All data generated or analysed during this study are included in the manuscript and supporting files.

The following datasets were generated:

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
