## [Decision Letter]

Thank you for submitting your article "NMDA spikes mediate amplification of odor pathway information in the piriform cortex" for consideration by *eLife*. Your article has been reviewed by three peer reviewers, and the evaluation has been overseen by a Reviewing Editor and a Senior Editor. The reviewers have opted to remain anonymous.

The reviewers have discussed the reviews with one another and the Reviewing Editor has drafted this decision to help you prepare a revised submission.

Summary:

Dendrites had been thought to be passive cables. Important work in other brain areas over the past few decades, including that from Schiller and Mel, has revealed that dendrites are often active, are compartmental, and can support both Na and NMDA spikes, all of which may increase their computational power. If dendrites in piriform cortex (PCx) pyramidal cells were similarly active, it could affect how odor information is integrated and transformed in the PCx. However, the only previous study to examine this question directly (Bathellier et al., 2009) concluded that these dendrites were largely passive, resulted in a simple, linear summation of inputs across the dendritic arbor.

Kumar et al. now revisit this question. Recording in acute rat brain slices, they can reliably evoke NMDA spikes (and sometimes Na spikelets) in PCx pyramidal cell dendrites using either glutamate uncaging or focal electrical stimulation of either lateral olfactory tract (LOT) or intra-cortical (IC) inputs. Computational studies support the sufficiency of their conclusions. This result may have important implications for understanding how odor information is integrated in PCx. It appears that both the Bathellier and the present works are done well, and it remains unclear how to resolve the discrepancy.

Although all the reviewers found this work to be potentially very interesting, they raised a number of concerns that the authors need to address before publication of this work in *eLife*.

Essential revisions:

1) Discrepancy between the present and the Bathellier paper. All three reviewers raised the issue why the present authors see NMDA spikes in the PCx and others did not. Although the reviewers thought that it is not required to fully explain the difference, it warrants more careful discussion.

Figure 1 shows that NMDA spikes can be evoked by uncaging, which has been little used in PCx, but most of the paper uses a very standard electrical stimulation protocol (50 Hz train of 3 weak stimuli, <20 µA) to show very clear regenerative responses at the soma. Many papers have used a similar train protocol in PCx (not just Bathelier et al.) with no sign of this strong somatic response.

Although the authors suggest that the discrepant results are due to a difference in the orientation of slice sectioning, this argument is not very convincing. The parasagittal cutting angle of Bathellier et al. (2009) preserves the physiologically relevant LOT and piriform architecture that is being examined in the current study. Cutting coronally severs physiological LOT inputs, and it is unclear why this cutting angle was chosen for the current study, given this disadvantage/complication. Did the authors perform experiments with the slice orientation of Bathellier et al.? Showing data recorded with slices in the Bathellier et al. orientation would significantly strengthen the manuscript.

There are other clear differences between their results and the results of Bathellier and co-workers that should be discussed. For example, the off kinetics after each stimulation in the stimulation train appear to differ markedly (see Figure 7 of Bathellier et al.). Did the authors ever find fast off kinetics? Also, the intracellular solutions are different. Please reconsider the evaluation of differences in the two protocols. Would stimulating LOT fibers in a more physiologically relevant manner (as in Bathellier et al.) reveal similar NMDA spikes in the pyramidal neurons?

It is not enough to mention their use of coronal slices (subsection “Mismatch to the currently accepted view of olfactory pyramidal neurons”) because many other papers also use coronal slices. Is it something about the age of the mice (a bit older than usual) or the recording temperature (a bit higher than usual)? If so, do these responses disappear with younger mice at lower temperatures?

Incidentally, it is puzzling that the authors observe occasional spontaneous large events (Figure 2), which are rarely seen in healthy slices but are sometimes seen in compromised tissue (which might be the case for older slices at warmer temperatures). Could NMDA spikes be related to quasi epileptiform activity in compromised tissue? Recall that PCx is highly epileptogenic because of its profuse associational (intracortical) connections, and network hyperexcitability might be confused with intrinsic electrogenesis (i.e. with NMDA spikes).

Pyramidal cells have postsynaptic GABA-B receptors. Did baclofen alter the cells' resting membrane potential? If so, surely this alone would directly alter the dendritic excitability independent of any effects on blocking transmitter release from IC inputs.

In the Materials and methods, the authors say that bicuculline was used in some of the experiments. The authors should be more specific about which experiments were done with inhibition blocked or intact, and whether or not that made any difference.

We hope that some of the above issues be addressed experimentally. If the authors choose not to perform further experiments, these issues need to be discussed more carefully.

2) Missing information about cell types: The paper is about "pyramidal cells" in the PCx but many of the figures do not show pyramidal cells. There are at least 4 main types of glutamatergic principal cells in PCx: layer 2a semilunar (SL), layer 2b superficial pyramidal (SP) cells, and layer 3 deep pyramidal and multipolar cells. The anatomical, circuit and intrinsic electrical properties of these different classes have been studied by many investigators over many years and the authors need to respect this previous work. It is also important to their interpretation. For example, the SL cells shown in Figures 2A, 3A and 4D have been shown by a number of groups to receive very weak intracortical inputs on their proximal dendrites, which means that claims of NMDA spikes triggered by proximal stimulation of this cell type could not be believed. Were any layer 3 principal cells studied? These are different again. The data need to be reanalyzed with careful consideration of the cell type, determined from the soma location and dendritic morphology.

3) Dendritic crosstalk experiments are hard to interpret: This is a key part of the paper that the authors use to infer function, but there are concerns. There seems to be an assumption that "beams" of axons can be separately stimulated within the LOT/layer 1a or within associational layers, like for Schaffer collaterals in CA1, but anatomical studies have shown that axons within the afferent and association layers of PCx do not travel straight and are highly branched (e.g. Haberly et al., 1978, 1984, 2001, etc). Thus, it is hard to accept that "local" extracellular stimulation can be used to separately activate two LOT inputs, or specifically activate sister branches, or selectively activate more proximal intracortical synapses by moving the stimulating electrode closer to the soma. The analysis of these data is also unclear. The idea of a constant "bias" input is used to look for threshold shifts vs. vertical shifts in the input-output plots and infer non-linear interactions. It is stated that threshold shifts are seen, but some plots show a mixture, not discussed (e.g. Figures 4G, 5C, 6C, 6E). If the two inputs are simply recruiting overlapping synapses with variable degrees of overlap, one might expect a complex picture of the kind that is seen. It is also a concern that "sister branches" and "different branches" are defined so loosely, without quantification, especially when noting that SL and SP cells have very different patterns of dendritic branching yet these cell types were not distinguished.

4) The authors allude to the fact that LOT inputs arrive first followed by IC inputs (subsection “NMDA spikes can be generated throughout the apical tree, and by both LOT and IC inputs”). However, they only examine simultaneous stimulation of different dendritic branches. What happens if they temporally offset, for example, stimulation of LOT and IC inputs? For example, Franks et al. (2011) showed the effectiveness of pairing subthreshold LOT and IC inputs depended on the order in which they were activated, which they attributed to recruiting feedback inhibition. Might dendritic non-linearities partially underlie this effect as well?

5) Functional importance: The authors hypothesize that clustered synaptic inputs can drive NMDA spikes that powerfully drive the cell under in vivo-like conditions, producing discontinuous receptive fields (subsection “Modeling”). The first part of this hypothesis is reasonable, based on the precedent from other cortical areas, although it is not convincing that the model replicates 'in vivo' conditions. Because of this, how these dendritic events alter somatic spiking (i.e. output) remains unclear.

For example, PCx neurons in vivo are constantly bombarded with synaptic inputs, depolarizing them well above the -70 mV found in the model. Also, little is actually known about inhibitory inputs in vivo, despite what is stated earlier; most of the supporting citations here refer to slice experiments. Thus, conclusions from the model are expected to be highly approximate. Furthermore, the connection between NMDA spikes and discontinuous receptive fields is not strongly argued. It is widely thought in the field that circuit phenomena, like recurrent inhibition, are more likely to be important than the cell-autonomous effects proposed in this paper. While it has been reported that baclofen blocks associational inputs but leaves the receptive field properties intact, baclofen is a dirty drug in this context (e.g. it also suppresses inhibition and causes epilepsy in PCx in vivo) so the authors should be wary about reading too much into this finding.

6) Figure 2A is described as showing focused stimulation of LOT fibers eliciting evoked posy-synaptic potentials recorded in the soma, and this is backed up by a decrease of the evoked potentials by APV. But, was there direct stimulation of the dendrite? Was the evoked potential completely blocked by inhibition with APV and CNQX (as Apicella et al., 2010)? Is this really a "synaptic" stimulation as stated in subsection “Initiation of NMDA spikes by activation of LOT inputs using focal synaptic stimulation.”?

7) Ca imaging in dendrites provides important information for the argument of local NMDA spikes. However, did the authors record from neighboring dendritic trees? Do these data demonstrate the independence of local dendrite responses?

8) What was the criterion used to determine whether stimulation was in the LOT vs the cortical input? I assume it was the DIC image. Please provide an image.

9) A t-test cannot be used for tests of significance between more than two samples (e.g. Figure 1D). Either use ANOVA if the assumption of normality is met, or a non-parametric Kruskal-Wallis (KW) ANOVA with post-hoc tests of between sample significance or use t-test corrected for multiple comparisons.

For example, in Figure 5E the "sister" and "different" branches were each compared with "same", but it was concluded that "threshold coupling between branches was weaker still when the bias was delivered to a different branch", i.e. a conclusion is now being made about differences between "sister" and "different" branches, which was not tested. This kind of statement requires a multivariate statistical test, e.g. ANOVA. A great deal of the Discussion draws upon results of this kind so it is important.

Along these same lines, the authors claim there is a significant interaction effect for co-activation of two LOT inputs (subsection “Combination selectivity and compartmentalization of pyramidal neuron dendrites”) on a single dendrite, yet no statistical assessment of an interaction (ANOVA, KW-ANOVA) was performed. These data should be analyzed properly with F-scores and degrees of freedom listed in the Results section and figure legends.

10) In the “Statistical procedure” subsection, the authors state that sample size was chosen based on "standards used in the field as well as our own vast experience…" While it is acceptable to use common standards in the field (though a power analysis based on previous results would be preferred), it is inappropriate to appeal to authority ("vast experience") in lieu of proper statistical procedures. Please remove that appeal to authority from the statistics section.

[Editors' note: further revisions were requested prior to acceptance, as described below.]

Thank you for submitting your article "NMDA spikes mediate amplification of inputs in the rat piriform cortex" for consideration by *eLife*. Your article has been reviewed by three peer reviewers, and the evaluation has been overseen by a Reviewing Editor and Catherine Dulac as the Senior Editor. The reviewers have opted to remain anonymous.

The reviewers have discussed the reviews with one another and the Reviewing Editor has drafted this decision to help you prepare a revised submission.

Specifically, all the reviewers found this work of importance and are positive for publication of your work at *eLife*. The revision has addressed almost all of their previous concerns. However, there are a few points to which we would like to see further revisions. For example, we would like you to address the concern regarding the cell types of investigation (Reviewer 2, the second paragraph).

*Reviewer #1:*

The authors have satisfactorily addressed all my substantive concerns.

Discussion section.

*Reviewer #2:*

The authors have satisfactorily addressed many of my criticisms, although I would like to place on record that I'm still concerned about the discrepancy with Bathelier et al. (2009). I note the authors' comment that the burden of proof should be on Bathelier et al. to come up with an explanation, but a general principle is that authors should aim to show both why they are right and why everybody else is wrong.

Point #2 about cell types is still a concern. Semilunar cells do not necessarily lack basal dendrites altogether (see Choy et al., 2017, who report that SL cells can have basal dendrites). The cell shown in Figure 1A appears to have its soma located in layer 2a, which is where SL cells are located, and looks very like the SL cells reported by Choy et al. (2017). I suggest the authors say they studied principal cells located in layer 2 but concentrated on superficial pyramidal cells. Saying that would be accurate, would avoid the possibility of raised eyebrows, and would not detract from the paper in any way.

*Reviewer #3:*

Thorough revision of the manuscript.

[Editors' note: further revisions were requested prior to acceptance, as described below.]

Thank you for submitting your article "NMDA spikes mediate amplification of inputs in the rat piriform cortex" for consideration by *eLife*. Your article has been reviewed by one peer reviewer, and the evaluation has been overseen by a Reviewing Editor and Catherine Dulac as the Senior Editor. The reviewer has opted to remain anonymous.

The Reviewing Editor has drafted this decision to help you prepare a revised submission.

The authors have addressed most of the previous concerns. However, there is a remaining issue regarding the authors' descriptions of the cell types studied. The reviewers think that further revisions are required to accurately report the results.

*Reviewer #2:*

I suggested that the authors state that they studied principal cells located in layer 2, mainly focusing on superficial pyramidal cells. I note that the term "pyramidal cells" is deeply embedded in their manuscript and it's unlikely they will change this now, especially given that a number of other groups in the field use a similar terminology. Hence, I'm prepared to overlook this. However, the authors must at least correct the statement in the Results:

"All our recorded neurons were pyramidal neurons from layer 2, mainly in the upper layer as determined by the Dodt contrast image and somatic firing pattern."

This is simply untrue. The cell shown in Figure 1A is not a pyramidal neuron, for multiple reasons noted previously. Pyramidal neurons are not located in the upper layer (presumably meaning upper layer 2?), as confirmed by multiple authors. And the neurons shown in Figure 2A and Figure 9A clearly do not have their somas located in upper layer 2 – instead, they are in deep layer 2 or layer 3. This sentence needs to be changed to:

"The majority of our recorded neurons were pyramidal neurons from layer 2, as determined from the Dodt contrast image and somatic firing pattern."

By saying this, the authors avoid stating an untruth at the very start, and at least gloss over the point that neuroscientists from Cajal onward have recognised that the PCx contains more than just one type of "pyramidal" principal neuron.

References:

Choy JMC, Suzuki N, Shima Y, Budisantoso T, Nelson SB, Bekkers JM. Optogenetic Mapping of Intracortical Circuits Originating from Semilunar Cells in the Piriform Cortex, Cerebral Cortex, Volume 27, Issue 1, 1 January 2017, Pages 589–601, doi: 10.1093/cercor/bhv258

---

## [Author Response]

Essential revisions:1) Discrepancy between the present and the Bathellier paper. All three reviewers raised the issue why the present authors see NMDA spikes in the PCx and others did not. Although the reviewers thought that it is not required to fully explain the difference, it warrants more careful discussion.Figure 1 shows that NMDA spikes can be evoked by uncaging, which has been little used in PCx, but most of the paper uses a very standard electrical stimulation protocol (50 Hz train of 3 weak stimuli, <20 µA) to show very clear regenerative responses at the soma. Many papers have used a similar train protocol in PCx (not just Bathelier et al.) with no sign of this strong somatic response.Although the authors suggest that the discrepant results are due to a difference in the orientation of slice sectioning, this argument is not very convincing. The parasagittal cutting angle of Bathellier et al. (2009) preserves the physiologically relevant LOT and piriform architecture that is being examined in the current study. Cutting coronally severs physiological LOT inputs, and it is unclear why this cutting angle was chosen for the current study, given this disadvantage/complication. Did the authors perform experiments with the slice orientation of Bathellier et al.? Showing data recorded with slices in the Bathellier et al. orientation would significantly strengthen the manuscript.There are other clear differences between their results and the results of Bathellier and co-workers that should be discussed. For example, the off kinetics after each stimulation in the stimulation train appear to differ markedly (see Figure 7 of Bathellier et al.). Did the authors ever find fast off kinetics? Also, the intracellular solutions are different. Please reconsider the evaluation of differences in the two protocols. Would stimulating LOT fibers in a more physiologically relevant manner (as in Bathellier et al.) reveal similar NMDA spikes in the pyramidal neurons?It is not enough to mention their use of coronal slices (subsection “Mismatch to the currently accepted view of olfactory pyramidal neurons”) because many other papers also use coronal slices. Is it something about the age of the mice (a bit older than usual) or the recording temperature (a bit higher than usual)? If so, do these responses disappear with younger mice at lower temperatures?Incidentally, it is puzzling that the authors observe occasional spontaneous large events (Figure 2), which are rarely seen in healthy slices but are sometimes seen in compromised tissue (which might be the case for older slices at warmer temperatures). Could NMDA spikes be related to quasi epileptiform activity in compromised tissue? Recall that PCx is highly epileptogenic because of its profuse associational (intracortical) connections, and network hyperexcitability might be confused with intrinsic electrogenesis (i.e. with NMDA spikes).Pyramidal cells have postsynaptic GABA-B receptors. Did baclofen alter the cells' resting membrane potential? If so, surely this alone would directly alter the dendritic excitability independent of any effects on blocking transmitter release from IC inputs.In the Materials and methods, the authors say that bicuculline was used in some of the experiments. The authors should be more specific about which experiments were done with inhibition blocked or intact, and whether or not that made any difference.We hope that some of the above issues be addressed experimentally. If the authors choose not to perform further experiments, these issues need to be discussed more carefully.

A major concerned raised is the discrepancy between the Bathellier paper and the results presented in this manuscript with regard to the regenerativity capabilities of PCx pyramidal neurons dendrites. We observed both sodium spikelet’s and NMDA spikes while the Bathellier paper failed to observe these regenerative events. Here are our answers to the various issues raised in Point #1:

a) Additional experiments with similar experimental conditions as in Bathellier paper. To address the reviewers’ concerns head on, we performed additional experiments using sagittal slices, identical intracellular solution (especially in light of the comment regarding the intracellular chloride concentration, see below) and temperature (see next point). Using the Bathellier slice conditions it was easily possible to evoke NMDA spikes in PCx pyramidal neurons. These spikes were robust, completely blocked by the NMDAR blocker APV. The residual synaptic response was further completely blocked with the AMPAR blocker CNQX. We show that these NMDA spikes evoke local calcium transients in the stimulated branch and fail to propagate to sister branches as it could not evoke significant calcium transients in sister unstimulated branches. These results are now shown in new Figure 2—figure supplement 1.

b) Recording temperature. The reviewer commented about the temperature which is stated to be 36°C in our experimental conditions. Upon more precisely measuring the bath temperature we noticed that our routine *bath* temperature is 34°C, 36°C was measured more distally at the tube. Thus the temperature in all our experiments is similar to that used by Bathellier et al. This is now corrected in the Materials and method section. With regard to the age of the animals used, since we did not perform experiments on young animals we cannot comment whether NMDA spikes can be initiated at a younger age in PCx pyramidal neurons.

c) Intracellular chloride concentration. The reviewer raises the question of intracellular chloride concentration we used and its physiological relevance. In our experiments we used “low chloride” concentration (4 mM) while Bathellier et al. used “high chloride” (20 mM). The question of what is the intracellular physiological chloride concentration is complex as chloride concentration is actively regulated. However, several groups have directly measured the intracellular chloride concentration in cortical neurons. For example, Boffi et al. (2018) measured an intracellular chloride concentration of 6 ± 2 mM in anesthetized and awake mice using Cl^-^ indicator with two photon microscopy. (See also Price and Trussell, 2006; Berglum et al., 2008.) Note that in the past our lab has used a “high chloride” intracellular concentration; we moved to lower concentrations as we believe that it is more physiological.

d) Spontaneous NMDA spikes. The reviewer commented about the occurrence of spontaneous NMDA spikes and the possibility this might be a sign of compromised slices. We note that spontaneous NMDA spikes occur rarely during an experiment. The resting membrane potential does not show any depolarization and we did not observe spontaneous firing of action potentials in neurons, thus it is unlikely that these spontaneous dendritic events represented an epileptic state. We should stress that using calcium imaging we are able to show that NMDA spikes are very localized to a dendritic branch supporting an isolated activation rather an epileptic-like state. We think that these spontaneous NMDA spikes may represent activation of a small connected subnetwork in the slice, which provides the physiological “substrate” for NMDA spike generation in vivo. Also, an important control that establishes that the regenerativity we describe is a local spike, and not due to recurrent epileptic network effects, is that these spikes are blocked with intracellular NMDAR blocker MK801 (Figure 2E).

e) Resting membrane potential. The editor commented about the possibility that the resting membrane potential may change with baclofen. The average resting membrane potential without baclofen is -75.51 ± 2.77 and after addition of baclofen is -75.77 ± 2.74 (p>0.5). This comparison has been added to the revised manuscript in subsection “NMDA spikes can be generated throughout the apical tree, and by both LOT and IC inputs”. It is also important to note that baclofen was used in a small subset of experiments (only for Figure 4D-H). We observed NMDA spikes without addition of baclofen both with synaptic stimulation and with glutamate uncaging. Thus it is unlikely that changes in resting membrane potential or any other nonspecific effects of baclofen are the cause of NMDA spike initiation.

f) Bicuculline. We now mention more clearly in the Materials and method section when bicuculline was used.

g) Discussion in the manuscript. As our new experiments (Figure 2—figure supplement 1) have ruled out some obvious possible factors such as slicing and solution composition conditions, we are left with speculative explanations as to the difference between our results and those of Bathellier et al. One difference we note is a more distributed recruitment of axons with their synaptic stimulation compared to ours, which is evident by the large AMPA responses in their recordings (see their Figure 6). We feel that a positive result obtained under standard conditions needs less explanation, rather the burden of proof should be passed to the group that reported negative results. With regard to other labs not reporting these spikes, actually we believe that the underlying supralinear summation seen by Davison and Ehlers (2011), for example their Figure 6D, and by Apicella et al. (2010) is probably related to NMDA spike initiation in dendrites of pyramidal neurons. This possibility is raised in the manuscript in the Discussion section.

2) Missing information about cell types: The paper is about "pyramidal cells" in the PCx but many of the figures do not show pyramidal cells. There are at least 4 main types of glutamatergic principal cells in PCx: layer 2a semilunar (SL), layer 2b superficial pyramidal (SP) cells, and layer 3 deep pyramidal and multipolar cells. The anatomical, circuit and intrinsic electrical properties of these different classes have been studied by many investigators over many years and the authors need to respect this previous work. It is also important to their interpretation. For example, the SL cells shown in Figures 2A, 3A and 4D have been shown by a number of groups to receive very weak intracortical inputs on their proximal dendrites, which means that claims of NMDA spikes triggered by proximal stimulation of this cell type could not be believed. Were any layer 3 principal cells studied? These are different again. The data need to be reanalyzed with careful consideration of the cell type, determined from the soma location and dendritic morphology.

We thank the reviewer for bringing up the issue of cell types in the context of piriform cortex. We are aware of the cell types mentioned by the reviewer. However, to reduce complexity, we intentionally chose to concentrate on the pyramidal neurons, leaving out the semilunar neurons. We disagree with the reviewer that the neurons shown on Figures 2A, 3A and 4D are semilunar. As pointed in Suzuki and Bekkers (2006), semilunar neurons are neurons found in the upper layer 2a and lack basal dendrites altogether. We apologize that the layer drawings were not accurate in Figure 2, and the reconstruction in Figure 3 was not elaborate enough with respect to the basal tree. However, all had basal dendrites and were not found in upper layer 2a. We remade the figures to highlight better the basal tree and corrected the layer drawing in Figure 1A and added the layer boundary for all neuron drawings where we have kept the DIC image. In line with Suzuki and Bekkers (2006), we also add somatic firing of the cells shown, to further support the claim that these are pyramidal neurons and not semilunar. Thus all of our neurons were either from layer IIA or layer IIB, and none were from layer 3 (this is now stated in the text).

As to the comment made by the reviewer regarding Figures 2A, 3A and 4D, in all of these cases stimulation was distal and presumably involved very few intracortical fibers. Thus even if they were semilunar (which they are not) this would not compromise our claims.

3) Dendritic crosstalk experiments are hard to interpret: This is a key part of the paper that the authors use to infer function, but there are concerns. There seems to be an assumption that "beams" of axons can be separately stimulated within the LOT/layer 1a or within associational layers, like for Schaffer collaterals in CA1, but anatomical studies have shown that axons within the afferent and association layers of PCx do not travel straight and are highly branched (e.g. Haberly et al., 1978, 1984, 2001, etc). Thus, it is hard to accept that "local" extracellular stimulation can be used to separately activate two LOT inputs, or specifically activate sister branches, or selectively activate more proximal intracortical synapses by moving the stimulating electrode closer to the soma. The analysis of these data is also unclear. The idea of a constant "bias" input is used to look for threshold shifts vs. vertical shifts in the input-output plots and infer non-linear interactions. It is stated that threshold shifts are seen, but some plots show a mixture, not discussed (e.g. Figures 4G, 5C, 6C, 6E). If the two inputs are simply recruiting overlapping synapses with variable degrees of overlap, one might expect a complex picture of the kind that is seen. It is also a concern that "sister branches" and "different branches" are defined so loosely, without quantification, especially when noting that SL and SP cells have very different patterns of dendritic branching yet these cell types were not distinguished.

The reviewer raises concerns that our stimulation might not be as focal as we think, thus leading to overlapping stimulation of different branches when we think we are stimulating branches separately. This in turn would cause a mixture of shifting and lifting of the input-output curves. We will try to answer this point in multiple ways:

a) We first mention evidence that our stimulation technique is for the most part focal. The technique of extracellular synaptic stimulation we use was specifically designed to produce a very concentrated current within a small volume. We use a double-barrelled electrode and pull the electrode to a final size of 2-3 µm. The two barrels serve as a bipolar electrode, thus constraining the current flow locally. With this method we are successful in stimulating only a small number of axons – those that are very close to the electrode. In previous projects we used calcium imaging to test the number of spines stimulated within a branch by this method of stimulation (Polsky et al., 2004; Sandler et al., 2016). We find that with this method we can activate a small dendritic segments with an average of 8 activated spines which run within 5 µm of our stimulating electrode. In this manuscript as well, the calcium imaging data strongly support our claim of focal stimulation (see Figure 2B and Figure 2—figure supplement 1).

b) Our data support the notion that individual dendrites of PNs in PCx integrate their inputs largely independently of each other, which is a signature feature of a 2-layer model. Theoretically, in a perfect 2-layer model with independent dendritic subunits, a constant bias on the same branch would lead to a clean left shift of the sigmoidal input-output curve, and a constant bias on a sister or any other branch would cause no shifting, but would instead lead to a pure lifting of the input-output curve, because the summation between the branches would be only at the soma. The clear and robust effect that we see, both in the individual experiments as well as the summary plots, is a much more pronounced shifting of the input-output curve for within-branch cases compared to sister or different branch cases (see Figure 5C and 6C-E). Thus, when a bias input is delivered to a sister or different branch, the i/o curves undergo much less shifting, and more uniform lifting. If there were crosstalk in the stimulation pattern, which is the reviewer’s concern, this would tend to *reduce* the difference between same branch and different branch cases, and correspondingly reduce the appearance of independence between dendrites. Thus, if crosstalk is occurring, then the level of independence we show is an underestimate of the actual degree of independence between the dendrites.

c) Specifically, to further address experimentally the issue, we conducted a new set experiments where we replaced one stimulating electrode with an uncaging spot. In these experiments we activated two LOT “inputs” one with focal synaptic stimulation and the other with uncaging within one branch, and later moved the synaptic stimulation to a different branch while keeping the uncaging spot in place. As is shown in a new Figure 5 —figure supplement 1 and in the text in subsection “Combination selectivity and compartmentalization of pyramidal neuron dendrites”, we observed a large spike threshold reduction with the synaptic bias that was delivered to the same branch, but not to the different branch (Threshold was lowered by 52.2 ± 2.34% and 15.6 ± 4.2% for bias in same or different branch respectively). These data are consistent with the results presented in Figure 5.

4) The authors allude to the fact that LOT inputs arrive first followed by IC inputs (subsection “NMDA spikes can be generated throughout the apical tree, and by both LOT and IC inputs”). However, they only examine simultaneous stimulation of different dendritic branches. What happens if they temporally offset, for example, stimulation of LOT and IC inputs? For example, Franks et al. (2011) showed the effectiveness of pairing subthreshold LOT and IC inputs depended on the order in which they were activated, which they attributed to recruiting feedback inhibition. Might dendritic non-linearities partially underlie this effect as well?

To test this interesting point experimentally will require a large set of data which we feel is beyond the scope of the present manuscript. To start teasing apart the mechanisms at the single-cell level, we used computer simulations which showed that strong supralinear LOT and IC input summation persisted when either of the inputs was temporally shifted by up to 20 ms. This is now described in Figure 8—figure supplement 3 and in the text in subsection “Modeling”. This result, which is in line with our previous findings (Polsky et al., 2004), suggests that although the supralinear integration time window is not entirely symmetrical, still a supralinearity can be obtained with either ordering. Simulations that include the effects of feedforward and feedback inhibition require many more parameters and assumptions, including at the network level. So as we say above regarding experiments, we feel that modeling the effects of inhibition on nonlinear dendritic integration in PCx is beyond the scope of the present study.

5) Functional importance: The authors hypothesize that clustered synaptic inputs can drive NMDA spikes that powerfully drive the cell under in vivo-like conditions, producing discontinuous receptive fields (subsection “Modeling”). The first part of this hypothesis is reasonable, based on the precedent from other cortical areas, although it is not convincing that the model replicates 'in vivo' conditions. Because of this, how these dendritic events alter somatic spiking (i.e. output) remains unclear.For example, PCx neurons in vivo are constantly bombarded with synaptic inputs, depolarizing them well above the -70 mV found in the model. Also, little is actually known about inhibitory inputs in vivo, despite what is stated earlier; most of the supporting citations here refer to slice experiments. Thus, conclusions from the model are expected to be highly approximate.

The question of whether olfactory receptive field can be discontinuous in a cell that is exposed to synaptic bombardment from cortical neurons is indeed highly interesting. To explore this issue, we modified our model. In its revised formulation, olfactory inputs are activated in conjunction with a background synaptic bombardment mediated by excitatory and inhibitory inputs. Because the actual cortical feedback onto principal cells is largely unknown, as mentioned by the reviewers, we decided to model a ‘difficult’ scenario in which this background conductance is large and highly variable (depolarizing the cells by ~11 mV). Our new results indicate that the synaptic integration rules we described before, still hold true even in this scenario. In addition, we now explicitly test for branch selective, discontinuous receptive fields in the model. We show that background excitatory and inhibitory bombardment doesn’t preclude the cell from detecting the spatial arrangement of active olfactory inputs. Postsynaptic depolarization was consistently and significantly larger when simulated glomerular inputs innervated the same distal apical branch compared to segregation of those input to two or more distinct dendrites. Thus, NMDA spikes can promote discontinuous receptive fields in a neuron that incorporates in vivo like background synaptic activity. This data is presented in new Figure 8 and is described in the text in subsection “Modeling”

Furthermore, the connection between NMDA spikes and discontinuous receptive fields is not strongly argued. It is widely thought in the field that circuit phenomena, like recurrent inhibition, are more likely to be important than the cell-autonomous effects proposed in this paper.

We agree that presently the field views discontinuous receptive fields as a purely circuit phenomena, but one reason for this is that a cell-autonomous mechanism of the kind we demonstrate here was supposedly ruled out by Bathellier et al. leaving a circuit-level mechanism – still undefined – as the accepted theory (even though it is inconsistent withDavison and Ehlers (2009) and Apicella et al. (2010) who showed that cortical feedback is not required for supralinear summation of glomerular outputs and odor responses.

By showing that nonlinear dendritic amplification mechanisms *can* support supralinear summation, our paper rules back in the possibility that cell-autonomous mechanisms could mediate discontinuous receptive fields in PCx. Of course, it is possible that both single-neuron and circuit-level mechanisms could play a role in establishing discontinuous RFs, but settling the question of the relative importance of each of these mechanisms is a matter to be taken up in future studies.

While it has been reported that baclofen blocks associational inputs but leaves the receptive field properties intact, baclofen is a dirty drug in this context (e.g. it also suppresses inhibition and causes epilepsy in PCx in vivo) so the authors should be wary about reading too much into this finding.

We agree that baclofen has multiple pharmacological effects. For example, Gerrard et al. (2018) demonstrated that activation of GABA-B receptors in vivo both suppresses and enhances activity in principal neurons, leading to hyper-depolarization due to a direct effect of baclofen on GABA-B receptors and epileptic bursts due to disinhibition of the network. However, in our experimental conditions, namely brain slices stimulated at sub-threshold intensities, we never observed any epileptiform-like discharges in any of our experiments. In our experimental conditions cells stayed subthreshold to action potential initiation. In the literature baclofen was used in multiple previous articles to distinguish LOT and intra-cortical inputs, and thus we felt compelled to repeat our experiments with this experimental paradigm. Our experimental results, of clean separation between the effects of Baclofen on the dendritic regions closer to the cell body, known to be mainly intra-cortical, and the lack of effect on synaptic signals in the distal regions of the LOT, is a clear indication of the validity of this approach.

6) Figure 2A is described as showing focused stimulation of LOT fibers eliciting evoked posy-synaptic potentials recorded in the soma, and this is backed up by a decrease of the evoked potentials by APV. But, was there direct stimulation of the dendrite? Was the evoked potential completely blocked by inhibition with APV and CNQX (as Apicella et al., 2010)? Is this really a "synaptic" stimulation as stated in subsection “Initiation of NMDA spikes by activation of LOT inputs using focal synaptic stimulation.”?

Following the reviewer comment we performed experiments to show that APV blocks the NMDA spikes but leave the AMPA synaptic potentials intact and further addition of CNQX blocks the AMPA synaptic potentials altogether (new Figure 2—figure supplement 1 and text in subsection “Initiation of NMDA spikes by activation of LOT inputs using focal synaptic stimulation.”).

7) Ca imaging in dendrites provides important information for the argument of local NMDA spikes. However, did the authors record from neighboring dendritic trees? Do these data demonstrate the independence of local dendrite responses?

As a routine we use calcium imaging to determine that our stimulation is focal as shown in Figure 2B. Also we measured the full calcium profile of an NMDA spike as shown in Figure 3. In response to the reviewer comment we also now show calcium imaging of EPSPs just subthreshold to NMDA spike generation, and NMDA spike evoked calcium transients in the stimulated and unstimulated dendrite. We find that using our focal synaptic stimulation method, we observe significant calcium transients only in stimulated branches. This is now shown in new (Figure 2—figure supplement 1 and text subsection “Initiation of NMDA spikes by activation of LOT inputs using focal synaptic stimulation.”.

8) What was the criterion used to determine whether stimulation was in the LOT vs the cortical input? I assume it was the DIC image. Please provide an image.

Yes, we used Dot gradient contrast video microscopy to identify the LOT region. An example is provided in the new Figure 2—figure supplement 1.

9) A t-test cannot be used for tests of significance between more than two samples (e.g. Figure 1D). Either use ANOVA if the assumption of normality is met, or a non-parametric Kruskal-Wallis (KW) ANOVA with post-hoc tests of between sample significance or use t-test corrected for multiple comparisons.For example, in Figure 5E the "sister" and "different" branches were each compared with "same", but it was concluded that "threshold coupling between branches was weaker still when the bias was delivered to a different branch", i.e. a conclusion is now being made about differences between "sister" and "different" branches, which was not tested. This kind of statement requires a multivariate statistical test, e.g. ANOVA. A great deal of the Discussion draws upon results of this kind so it is important.Along these same lines, the authors claim there is a significant interaction effect for co-activation of two LOT inputs (subsection “Combination selectivity and compartmentalization of pyramidal neuron dendrites”) on a single dendrite, yet no statistical assessment of an interaction (ANOVA, KW-ANOVA) was performed. These data should be analyzed properly with F-scores and degrees of freedom listed in the Results section and figure legends.

We thank the reviewer for pointing out these statistical issues. We now corrected the statistical tests as requested in the main text (subsection “Combination selectivity and compartmentalization of pyramidal neuron dendrites”) and figure legends (Figures 1 and 5). Statistical tests were performed with Prism 7 (Graphpad) software.

10) In the “Statistical procedure” subsection, the authors state that sample size was chosen based on "standards used in the field as well as our own vast experience…" While it is acceptable to use common standards in the field (though a power analysis based on previous results would be preferred), it is inappropriate to appeal to authority ("vast experience") in lieu of proper statistical procedures. Please remove that appeal to authority from the statistics section.

Corrected.

[Editors' note: further revisions were requested prior to acceptance, as described below.]

Specifically, all the reviewers found this work of importance and are positive for publication of your work at eLife. The revision has addressed almost all of their previous concerns. However, there are a few points to which we would like to see further revisions. For example, we would like you to address the concern regarding the cell types of investigation (Reviewer 2, the second paragraph).Reviewer #2:The authors have satisfactorily addressed many of my criticisms, although I would like to place on record that I'm still concerned about the discrepancy with Bathelier et al. (2009). I note the authors' comment that the burden of proof should be on Bathelier et al. to come up with an explanation, but a general principle is that authors should aim to show both why they are right and why everybody else is wrong.

We generally agree, we indeed made great experimental efforts to come up with an answer, but with no success.

Point #2 about cell types is still a concern. Semilunar cells do not necessarily lack basal dendrites altogether (see Choy et al., 2017, who report that SL cells can have basal dendrites). The cell shown in Figure 1A appears to have its soma located in layer 2a, which is where SL cells are located, and looks very like the SL cells reported by Choy et al. (2017). I suggest the authors say they studied principal cells located in layer 2 but concentrated on superficial pyramidal cells. Saying that would be accurate, would avoid the possibility of raised eyebrows, and would not detract from the paper in any way.

This is now corrected in the manuscript as suggested by the reviewer.

[Editors' note: further revisions were requested prior to acceptance, as described below.]

Reviewer #2:I suggested that the authors state that they studied principal cells located in layer 2, mainly focusing on superficial pyramidal cells. I note that the term "pyramidal cells" is deeply embedded in their manuscript and it's unlikely they will change this now, especially given that a number of other groups in the field use a similar terminology. Hence, I'm prepared to overlook this. However, the authors must at least correct the statement in the Results:"All our recorded neurons were pyramidal neurons from layer 2, mainly in the upper layer as determined by the Dodt contrast image and somatic firing pattern."This is simply untrue. The cell shown in Figure 1A is not a pyramidal neuron, for multiple reasons noted previously. Pyramidal neurons are not located in the upper layer (presumably meaning upper layer 2?), as confirmed by multiple authors. And the neurons shown in Figure 2A and Figure 9A clearly do not have their somas located in upper layer 2, instead, they are in deep layer 2 or layer 3. This sentence needs to be changed to:"The majority of our recorded neurons were pyramidal neurons from layer 2, as determined from the Dodt contrast image and somatic firing pattern."By saying this, the authors avoid stating an untruth at the very start, and at least gloss over the point that neuroscientists from Cajal onward have recognised that the PCx contains more than just one type of "pyramidal" principal neuron.

We have changed the sentence in the manuscript as suggested.

References:

Boffi JC, Knabbe J, Kaiser M, Kuner T. KCC2-dependent Steady-state Intracellular Chloride Concentration and pH in Cortical Layer 2/3 Neurons of Anesthetized and Awake Mice. 2018. Frontiers in Cellular Neuroscience. DOI: 10.3389/fncel.2018.00007

Gerrard LB, Tantirigama MLS, Bekkers JM. Pre- and Postsynaptic Activation of GABAB Receptors Modulates Principal Cell Excitation in the Piriform Cortex. 2018. Frontiers in Cellular Neuroscience. DOI: 10.3389/fncel.2018.00028

Price GD, Trussell LO. Estimate of the Chloride Concentration in a Central Glutamatergic Terminal: A Gramicidin Perforated-Patch Study on the Calyx of Held. 2006. Journal of Neuroscience. 26 (44) 11432-11436. DOI: 10.1523/JNEUROSCI.1660-06.2006